**Deeply subducted continental fragments: I. Fracturing, dissolution-precipitation and diffusion processes recorded by garnet textures of the central Sesia Zone (Western Italian Alps)**

Francesco Giuntoli[1], Pierre Lanari[1], Martin Engi[1]

[1]Institute of Geological Sciences, University of Bern, Baltzerstrasse 1+3, 3012 CH-Bern

*Correspondence to*: Francesco Giuntoli (francesco.giuntoli@plymouth.ac.uk)

**Abstract.** Contiguous continental high-pressure terranes in orogens offer insight into deep recycling and transformation processes that occur in subduction zones. These remain poorly understood, and currently debated ideas need testing. The approach we chose is to investigate in detail the record in suitable rocks samples that preserve textures and robust mineral
assemblages, which withstood overprinting during exhumation. We document complex garnet zoning in eclogitic micaschists from the Sesia Zone (Western Italian Alps). These retain evidence of two orogenic cycles and provide detailed insight into resorption, growth and diffusion processes induced by fluid pulses at high pressure conditions. We analysed local textures and garnet compositional patterns, which turned out remarkably complex. By combining these with thermodynamic modelling, we could unravel and quantify repeated fluid-rock interaction processes. Garnet shows low-Ca
porphyroclastic cores that were stable at (Permian) granulite facies conditions. The series of rims that surround these cores provides insight into the subsequent evolution: The first garnet rim that surrounds the pre-Alpine granulite facies core in one sample indicates that pre-Alpine amphibolite facies metamorphism followed the granulite facies event. In all samples documented, cores show lobate edges and preserve inner fractures, which are sealed by high-Ca garnet that reflect Alpine high-pressure conditions. These observations suggest that during early stages of subduction, before hydration of the
granulites, brittle failure of garnet occurred, indicating high strain rates which may be due to seismic failure. Several Alpine rims show conspicuous textures indicative of interaction with hydrous fluid: (a) resorption-dominated textures produced lobate edges, at the expense of the outer part of the granulite core; (b) peninsulas and atoll garnet are the result of replacement reactions; (c) spatially limited resorption and enhanced transport of elements due the fluid phase is evident along brittle fractures and in their immediate proximity. Thermodynamic modelling shows that all of these Alpine rims
formed at eclogite facies conditions. Structurally controlled samples allow these fluid-garnet interaction phenomena to be traced across a portion of the Sesia Zone, with a general decrease in fluid-garnet interaction observed towards the external, structurally lower parts of the terrane. Replacement of the Permian HT assemblages by hydrate-rich Alpine assemblages can reach nearly 100% of the rock volume. Since we found no clear relationship between discrete deformation structures (e.g. shear zones) observed in the field and the fluid pulses that triggered the transformation to eclogite facies assemblages, we
conclude that disperse fluid flow was responsible for the hydration.

**1 Introduction**

Unravelling the metamorphic and deformational history in polyorogenic complexes is challenging, as relics from previous stages commonly were partially or completely overprinted during the subsequent stages. Garnet is a robust and common mineral frequently used to decipher metamorphic processes. It grows in a wide range of P-T conditions and rock types (e.g. Spear et al., 1984;O'Brien, 1997).

However, various processes may alter zoning recorded by garnet, notably replacement reactions and intracrystalline diffusion. Replacement of garnet may alter or completely obliterate garnet growth zones due to dissolution and reprecipitation processes (Putnis, 2002, 2009;Putnis and John, 2010;Ague and Axler, 2016). Relics will show resorption features, e.g. lobate or peninsular structures within a garnet growth zone, and sharp transitions in composition may be observed (e.g. Cruz, 2011). If the entire centre of a garnet is replaced by other mineral, typical atoll garnet may form (Atherton and Edmunds, 1966;Cooper, 1972;Smellie, 1974;Homam, 2003;Cheng et al., 2007;Faryad et al., 2010;Cruz, 2011;Ortolano et al., 2014a). Dissolution of garnet is thought to be linked to the presence of a (reactive) fluid, and may be followed by precipitation of new garnet with a different composition (Hames and Menard, 1993;Compagnoni and Hirajima, 2001;Cheng et al., 2007;Faryad et al., 2010;Wassmann and Stöckhert, 2013;Ortolano et al., 2014a;Ague and Axler, 2016).

Diffusivity depends on temperature, time, and garnet composition, being generally higher for $Fe^{2+}$, Mg and Mn than for Ca (Carlson, 2006). Intracrystalline diffusion is relatively slow up to 700°C (e.g. Yardley, 1977;Spear, 1991;Carlson and Gordon, 2004;Caddick et al., 2010;Ague and Carlson, 2013): Below 700°C, garnet of a few hundred micrometres radius will retain compositional zoning for several tens of million years (Florence and Spear, 1991;Caddick et al., 2010).

Reconstructing the polyorogenic history stored in garnet requires an understanding of both texture and compositional zoning. To unravel the information recorded by garnet, we combined high-resolution electron probe compositional mapping with forward thermodynamic modelling. Our models account for fractional crystallisation and garnet resorption (Lanari et al., 2017). We applied this approach to several samples collected from the Internal Complex (as defined in Giuntoli and Engi, 2016) of the Sesia Zone, where garnet retains information from two orogenic cycles at very different conditions: Permian high-temperature followed by Alpine high-pressure (details in the next section).

We show that the textural and compositional changes are related to processes of dissolution, growth and diffusion. We discuss quantitative pressure (P) and temperature (T) information of garnet growth zones (data from Lanari et al., 2017; Giuntoli et al., in review) in relation with the observed textures. We propose that repeated local resorption and growth of garnet account for the complexly zoned garnets with lobate textures, peninsular features and, in the most extreme cases, formation of atoll garnet. Localized intracrystalline diffusion is observed only in the proximity of fractures, which were probably acting as main fluid pathways. For a series of four samples collected 2 kilometres across the main foliation and 5 kilometres along strike of the Internal Complex of the Sesia Zone, we infer that these garnet resorption and replacement processes are related to repeated fluid-rock interaction.

## 2 Geological setting and previous work on garnet

The Alpine orogenic belt formed between the Cretaceous and the Oligocene as a result of the convergence of the European plate and the Adriatic plate (Dewey et al., 1989;Rosenbaum et al., 2002;Handy et al., 2010). The Sesia Zone is a major high-pressure continental terrane located in the Italian Western Alps (Fig. 1a). It is thought to derive from the distal portion of the Adriatic passive margin that was involved in the Alpine subduction (Dal Piaz, 1999;Beltrando et al., 2014). The Sesia Zone is bounded to the west by blueschist to eclogite facies subunits derived from the Piemontese-Liguria Ocean (Bearth, 1967;Dal Piaz and Ernst, 1978;Martin et al., 1994;Cartwright and Barnicoat, 2002;Bucher et al., 2005;Groppo et al., 2009;Rebay et al., 2012;Negro et al., 2013) and to the east by the Insubric Line. This dextral brittle fault separates the Sesia Zone from the Southern Alps, which show a weak Alpine imprint at sub-greenschist facies (Bertolani, 1959;Zingg, 1983).

In the Aosta Valley (Fig. 1b) the Sesia Zone comprises two main complexes, i.e. the Internal and External Complexes (IC and EC hereafter; Giuntoli and Engi, 2016), separated by a greenschist facies shear zone. The EC comprises three epidote blueschist facies sheets of orthogneiss, with only minor paragneiss and metasediments, which are separated by lenses and bands, classically named Seconda Zona Diorito-Kinzigitica (2DK; Artini and Melzi, 1900;Dal Piaz et al., 1971;Compagnoni et al., 1977); these retain a pre-Alpine high-temperature imprint and only local evidence of the Alpine HP-history. The IC consists of several eclogitic subunits, each 0.5–3 km thick, separated by presumed monometamorphic (Mesozoic) meta-sediments (Venturini et al., 1994;Regis et al., 2014;Giuntoli and Engi, 2016). This complex is characterized by interlayered micaschist, eclogite, ortho- and paragneiss, with a very minor portion of pre-Alpine marbles and Mesozoic metasedimentary bands (e.g. Compagnoni, 1977;Castelli, 1991). The IC experienced eclogite facies conditions during Alpine metamorphism, with maximum recorded pressure of 2 GPa and temperature of 650-670 °C, between 85 and 55 Ma (Compagnoni, 1977;Konrad-Schmolke et al., 2011;Rubatto et al., 2011;Regis et al., 2014).  Minor and local retrograde blueschist and greenschist facies overprints are present, especially close to tectonic contacts (Babist et al., 2006;Giuntoli and Engi, 2016). The IC preserves rare relics of pre-Alpine (Permian) granulite facies conditions (0.6-0.9GPa, ~850°C) and very sparse retrograde amphibolite facies conditions (0.5-0.3 GPa, 570-670°C; Lardeaux and Spalla, 1991;Rebay and Spalla, 2001;Zucali et al., 2002).

Polymetamorphic pre-Alpine garnet has been recognized in the IC since the works of Martinotti (1970), Compagnoni (1977), Zucali et al. (2002), and Robyr et al. (2014). This type of garnet shows coarse, optically bright pre-Alpine porphyroclasts, which are generally mantled by a rim of Alpine garnet with a dusty appearance due to inclusions. Alpine garnet is also present as smaller euhedral crystals. During greenschist-facies Alpine retrogression, garnet grains were partially retrogressed to chlorite, especially along grain boundaries and fractures. Robyr et al. (2014) found that the garnet texture depends on the matrix: Garnet forms mushroom- and atoll-shaped crystals in quartz-rich layers, but large idioblastic crystals in mica-rich layers. Four growth zones were identified, the first three of which were modelled as stable during prograde pre-Alpine Barrovian metamorphism from 500 to 600°C and 0.6 to 0.9 GPa. The fourth zone was modelled as stable at 550°C and 1.7-2.2 GPa, hence it is certainly Alpine in age, as pre-Alpine metamorphism had not reached eclogite facies conditions. Similar

garnet zoning was interpreted differently by Konrad-Schmolke et al. (2006) who concluded that the zones do not reflect two orogenic cycles, but are related to different water contents of the protoliths during Alpine HP subduction. In water-saturated rocks garnet would then display typical prograde zoning; whereas in water-undersaturated rocks garnet would produce more complex zoning, including abrupt compositional changes from core to rim.

## 3 Analytical methods

Numerous thin sections were studied by optical microscopy to characterize the structural and metamorphic relations among the main mineral phases. Backscattered electron (BSE) images were obtained using a Zeiss EVO50 scanning electron microscope at the Institute of Geological Sciences, University of Bern, using an accelerating voltage from 15 to 25 keV, a beam current of 500 pA, and a working distance of 10 mm.

Electron probe micro-analyses (EPMA) were performed using a JEOL JXA-8200 superprobe at the Institute of Geological Sciences, University of Bern. For compositional mapping, the procedure of Lanari et al. (2013) was used. Spot analyses were measured for each mineral phase present in the area of the maps before the maps were acquired. The investigated areas were mapped in wavelength-dispersive mode (WDS), with point analyses serving as internal standards. For point analyses, the analytical conditions were 15 KeV accelerating voltage, 20 nA specimen current, 40 s dwell times (including $2 \times 10$ s of background measurement), and ~1 µm beam diameter. Nine oxide compositions were analysed, using synthetic and natural standards: wollastonite/almandine ($SiO_2$), almandine ($Al_2O_3$), anorthite ($CaO$), almandine ($FeO$), spinel ($MgO$), orthoclase ($K_2O$), albite ($Na_2O$), ilmenite ($TiO_2$), and tephroite ($MnO$). For X-ray maps, analytical conditions were 15 KeV accelerating voltage, 100 nA specimen current, dwell times of 150-250 ms, and step sizes from 3 to 5 µm. Nine elements (Si, Ti, Al, Fe, Mn, Mg, Na, Ca and K) were measured at the specific wavelength in two passes. Intensity maps were standardized using spot analyses as internal standards. X-ray maps were classified and standardized using XMAPTOOLS 2.2.1 (Lanari et al., 2014). Structural formulae and end-member proportion maps were generated using the external functions provided in XMAPTOOLS. The following garnet end-members were used: Grossular ($Ca_3Al_2Si_3O_{12}$), Almandine ($Fe_3Al_2Si_3O_{12}$), Pyrope ($Mg_3Al_2Si_3O_{12}$), Spessartine ($Mn_3Al_2Si_3O_{12}$). Major element compositions were analysed by X-ray fluorescence (XRF) spectrometry at the University of Lausanne (Switzerland). Representative amounts of the samples were crushed and then pulverized in a tungsten carbide mill. The powder was dried for two hours at 105°C. Loss of ignition was then determined by weight difference after heating to 1050°C for 3 hours.

## 4 Sample selection strategy and petrography

The samples investigated are pelitic micaschists of the IC of the Sesia Zone displaying pale orange to greyish weathering surfaces. Grain sizes range from sub-millimetric to a few centimetres, with garnet in all samples showing either a single or bimodal size range. In this paper the term large grains refers to garnet several millimetres in diameter, small grains are 50-

200 µm in diameter. In plane-polarized transmitted light large garnet shows a clear core surrounded by a first rim that is dark due to finely dispersed inclusions, mostly of rutile needles (5-20 µm; Fig. 2, 3). Adjacent to this cloudy rim, some garnet grains show a clear outermost rim. The same core-rim structure is visible in BSE images, in which the core is brighter than the rims (e.g. Fig. 6a). In detail, garnet shows slightly different textures and compositions in each sample; petrographic and microstructural characteristics are summarized below and in Table 1.

These petrographic features match observations reported by previous authors (see section 2) who attempted to distinguish Alpine from pre-Alpine garnet in the Sesia Zone. Based on these observations, the sample selection strategy we adopted in this study used the following criteria:

- The presence in hand-specimens of garnet of diverse sizes (sub-millimetric to a few centimetres);
- A bimodal distribution of garnet in the same thin section, with the larger crystals displaying a bright core and dark rims and smaller euhedral crystals (Fig. 2);
- Bright cores surrounded by darker rims visible in the SEM with a BSE detector (e.g. Fig. 6a).

This list of criteria was adopted to investigate the widest possible range of microtextures and processes recorded by garnet. Applying the criteria lead us to concentrate on only ~5% of all the samples taken, mostly because larger (pre-Alpine) garnet grains rarely survived in the area we studied. In fact, almost all of the pre-Alpine HT assemblages had re-equilibrated in hydrate-rich Alpine assemblages.

Four of the samples used were from internal parts (South East - FG1315 and FG12157) and external parts (North West-FG1249 and FG1347) of the IC (Fig. 1b). Major element compositions for the samples are shown in the Supplement Table S1. FG1315 is a garnet white mica schist with a pervasive foliation marked by phengite, paragonite and allanite (Fig. S1). It shows a banding of quartz-rich and mica-rich layers. A stretching lineation, marked by mica and quartz, is well visible in the field and was confirmed by optical microscopy: Quartz shows a crystallographic preferred orientation. Allanite grains are elongate in the main foliation; they show a rim of clinozoisite up to 20 µm thick; rare monazite relics are preserved in the core. Rutile, graphite and zircons are present as accessory phases. Garnet displays two different micro-textures: (a) large crystals preserve a porphyroclastic core plus overgrowth zones, and (b) atoll garnet a few hundred microns in size. Where the garnet core is preserved, it is optically clear and shows inclusions of rutile containing rare ilmenite relics. The first garnet rim appears cloudy due to finely dispersed rutile inclusions, the outer rim is clear with few and coarser rutile inclusions (Fig. S5b). Both rims contain abundant quartz inclusions. Atoll garnet is optically clear; its core is now mainly composed of quartz and sparse phengite. Brittle fractures in both garnet types are lined by minor late chlorite. Locally, chlorite, albite and clinozoisite mark limited greenschist retrogression.

FG12157 is a glaucophane garnet micaschist. Its foliation is deformed by open folds and marked by phengite, glaucophane, and allanite (Fig. S2). Microscopically, glaucophane shows two growth zones, with lighter cores and darker blue pleochroic rims due to higher Fe-content. Some crystals are also rimmed by green Ca-amphibole. Allanite shows a rim of clinozoisite, up to 20 µm thick. Chlorite, albite and green amphibole mark greenschist retrogression. Accessory phases are graphite, zircon, and rutile; the latter has a titanite overgrowth followed by an ilmenite rim. Garnet is present as large grains with a

clear core with several quartz inclusions at the periphery. The first garnet rim (Rim1) is dark, due to fine sagenitic rutile inclusions, locally as needles with a 120° intersection that may mark dissolved Ti-rich biotite. The second garnet rim (Rim2) is more clear (Fig. 3). Glaucophane and phengite locally occur within Rim1 or between Rim1 and Rim2. Apatite inclusions are presents in the core and both rims.

FG1249 is a rutile garnet micaschist with phengite, paragonite, allanite, and rutile defining an intense foliation that wraps around garnet of several millimetres size (Fig. S3). Allanite occurs rimmed by epidote; monazite is a rare relic in allanite cores. Sparse glaucophane is partly overgrown by albite and green amphibole. Garnet shows an optically clear core and a dark rim due to finely dispersed rutile (µm size) and paragonite inclusions. Coarser rutile (up to 100 µm) is included in the core as well as are phengite and apatite; paragonite and quartz are concentrated at the core-rim transition. Some chlorite fractures dissect entirely the garnet grains. Chlorite, albite, epidote, and green amphibole also reflect limited greenschist retrogression.

FG1347 is a chloritoid garnet micaschist with a strong foliation marked by phengite, chloritoid, paragonite, allanite and rutile that wraps around large garnet porphyroblasts. (Fig. S4). In the field an intense stretching lineation is marked by chloritoid. Microscopically, chloritoid is found in two generations, the younger of which is poikiloblastic and overgrows the main foliation. Accessory phases are zircon and opaque minerals. Some hundred microns monazite grains are preserved and are partially overgrown by allanite and apatite symplectites; allanite is rimmed by clinozoisite. Chlorite grew at the expense of the garnet rim and along brittle fractures. Garnet is several millimetres in size, with a clear core and a dark rim full of fine rutile inclusions, as in the other samples. Quartz inclusions are abundant in the rim (Fig. S6b).

## 5 Results

### 5.1 Evolution of successive garnet stages

High-resolution X-ray maps were helpful to study garnet microtextures, because element distributions were well visible and allowed us to investigate patterns that will be demonstrated to reflect growth and dissolution processes. The complex geometry of end-member proportion maps reveals several zones of distinct composition (Figs. 3-4) surrounding the garnet core, which are neither visible by optical microscopy nor in backscatter electron images. While some geometric variability is evident among individual grains and from one sample to another, certain prominent features are consistently observed in most of them. Maps of grossular fraction ($X_{Grs}$, Fig. 4) provide the clearest distinction of growth zones and are most suitable to identify similarities among all the samples. Generally, maps of the almandine and pyrope fractions provide patterns similar to grossular (except in one sample, FG1249, discussed below). The common feature evident in $X_{Grs}$ maps of all samples is a core poor in Ca, containing fractures sealed by a more calcic garnet. The core is surrounded by several rims, all of them with higher $X_{Grs}$ (Fig. 4), but subtle differences exist between the rims of each specimen, as discussed in the following sections.

Two types of fractures are visible: (i) millimeter-long fractures dissect entire garnet grains; these late fractures (Fig. 3c) are lined by chlorite; (ii) micrometer-thin cracks, visible only in compositional maps, form a fracture network in garnet cores; these so-called inner fractures were sealed by garnet richer in grossular (Fig. 3c; Fig. 4). Average compositions of garnet growth zones are reported in Table 2.

### 5.1.1 FG1315

The $X_{Grs}$ map shows a porphyroclastic core of uniform composition ($Alm_{68}Prp_{27}Grs_4Sps_1$), with lobate edges and a dense network of inner fractures sealed by more calcic garnet ($Alm_{67}Prp_{23}Grs_9Sps_1$; Figs. 4a, S5). Rim1 surrounds the rounded relic core and is higher in grossular content ($Alm_{60}Prp_{20}Grs_{19}Sps_1$) that the core; Rim1 is thinner in the direction of the main foliation, peninsular growth of Rim1 extends into the core. Rim2 ($Alm_{64}Prp_{24}Grs_{11}Sps_1$) is found as three textural types: (a) It grew externally onto Rim1; (b) between core and Rim1, and also surrounds the Rim1 peninsula, thus extending it (by ~20 µm); (c) fine veins (5-20 µm thick) dissecting Rim1 also show Rim2 composition (as discussed in section 6.2). The outermost rim (Rim3) is lower in grossular ($Alm_{69}Prp_{24}Grs_6Sps_1$); it displays peninsular growths inside Rim1 and Rim2. Remarkably, Rim3 is thin (~100 µm) parallel to the main foliation and thicker (~400 µm) perpendicular to it. Zoning patterns in atoll garnet (Fig. 5) are analogous, as are the grossular contents. The inner growth surfaces of Rim1 define negative garnet crystal forms, whereas Rim2 overgrowths are present on the outside growth surface only. Rim3 is the outermost growth zone, with peninsulas that formed at the expense of the previous rims.

### 5.1.2 FG12157

Garnet cores are chemically zoned, with domains showing high spessartine and pyrope contents but relatively low almandine fractions (Fig. 6; zoning profile of the garnet end-member in Fig. 2 of Lanari et al., 2017). The porphyroclastic core (average composition: $Alm_{69}Prp_{25}Grs_4Sps_2$) shows lobate edges and is fractured, but cracks are sealed by more calcic garnet ($Alm_{66}Prp_{21}Grs_{12}Sps_1$; Figs. 3c, 4b, 6c). Three overgrowth zones surround the core: Rim1 is higher in grossular ($Alm_{63}Prp_{20}Grs_{16}Sps_1$) than the core; Rim2 is higher yet ($Alm_{59}Prp_{17}Grs_{23}Sps_1$) and grew both internally and externally of Rim1. Rim3 is present just locally as the outermost rim ($Alm_{64}Prp_{20}Grs_{15}Sps_1$).

### 5.1.3 FG1249

A core and three rim generations are evident in the compositional maps (Fig. 7; zoning profile in Fig. 8). As in the other samples, the core is lowest in grossular ($Alm_{72}Prp_{18}Grs_5Sps_5$) and shows a fracture pattern, which was sealed by garnet with higher grossular contents ($Alm_{60}Prp_{18}Grs_{20}Sps_2$; Figs. 4c, 7c). Compared to the core, Rim1 is higher in grossular ($Alm_{75}Prp_{15}Grs_9Sps_1$) and its outer edge is euhedral. Rim2 is higher in grossular than the previous growth zone ($Alm_{62}Prp_{20}Grs_{17}Sps_1$) and quite similar to the fracture fillings. It is variable in thickness, but present both at the core-Rim1 boundary and peripheral to Rim1. Rim3 is the outermost growth zone and highest in grossular ($Alm_{58}Prp_{18}Grs_{23}Sps_1$). Its thickness varies, giving the entire garnet a euhedral shape. Locally, Rim3 is present also at the core-Rim1 boundary (Fig. 8).

This zoning pattern shows different features along the main fracture zones and around them: A sharp and strong decrease in Alm and Sps and increase in Prp contents are visible. Rim 1, 2 and 3 are also found in small garnet crystals (ø 0.1-0.5 mm) in which no core is visible.

### 5.1.4 FG1347

The core is zoned, with patchy areas high in $X_{Sps}$ and $X_{Alm}$ but low in $X_{Prp}$ (Fig. S6). The $X_{Grs}$ map (Fig. 4d) reveals a dense fracture pattern in the core ($Alm_{68}Prp_{27}Grs_3Sps_2$) sealed by more calcic garnet ($Alm_{68}Prp_{25}Grs_6Sps_1$). These fractures stop immediately against Rim1, which is highest in grossular content ($Alm_{66}Prp_{22}Grs_{11}Sps_1$). Rim2 ($Alm_{67}Prp_{26}Grs_6Sps_1$) is strongly asymmetric: It is wider perpendicular to the main foliation, as are Rim3 ($Alm_{69}Prp_{26}Grs_4Sps_1$) and Rim4 ($Alm_{71}Prp_{25}Grs_3Sps_1$). Additionally, clasts of Rim2, Rim3 and Rim4 are partly dismembered from the main garnet crystal, indicating deformation, which we note to be conspicuously absent in the other samples.

### 5.2 Modeling phase equilibria in partially re-equilibrated rocks

Several garnet growth zones were identified in each sample by detailed analysis of the end-member proportion maps using XMAPTOOLS (Lanari et al., 2014). Representative areas were selected and their average garnet composition extracted. Such areas are chemically homogeneous and uniform amongst different garnet grains, they display no evidence of enrichment or depletion in major elements. The quantitative micro-mapping strategy employed in this study has well established advantages (e.g. Marmo et al., 2002;Lanari et al., 2013;Ortolano et al., 2014b;Angiboust et al., 2016) over traditional spot analyses: (1) it allows key relationships, such as the successive growth zones, to be identified and relevant compositions to be constrained, (2) it permits testing if chemical zoning patterns are consistent over several grains, which helps support (or refute) the assumption of grain boundary equilibrium (e.g. Lanari and Engi, 2017), (3) it can be used to approximate local reactive bulk composition by accounting for (non-reactive) mineral relics. In all the samples of the present study, growth zone patterns and compositions of the mineral phases were consistent at thin section scale.

The complexity of garnet compositional zoning shown in Figures 4, 5, 6, 7 indicates that isochemical phase diagrams (or pseudosections) must be used with due caution. Previous studies have demonstrated that garnet fractionation potentially affects the reactive bulk composition (Evans, 2004;Robyr et al., 2014;Konrad-Schmolke et al., 2008) and thus shifts the calculated garnet isopleths in a P-T diagram (Lanari and Engi, 2017). However, garnet fractionation is not easy to account for where several growth stages are evident, as well as intermittent dissolution (which we show to be the case in the companion paper by Giuntoli et al., in review). Since the older growth zones are but partially preserved, as indicated e.g. by lobate edges (Figs. 3, 4, 6, 7) in our samples, a specific modelling strategy was developed and implemented in a computer program (GRTMOD: Lanari et al., 2017). In essence, for each garnet growth zone, the reactive bulk composition is optimized jointly with the P-T conditions to predict (using Theriak-Domino, de Capitani and Petrakakis, 2010) a garnet composition that matched the measured one. Results were accepted if the residual value (the least square sum of the differences) between the modelled and observed garnet compositions (end-member fractions) was <0.05, reflecting a close match. Previously

formed garnet generations were removed from the bulk rock composition, based on the analysis of end-member proportion maps. By analogy, in the case of garnet resorption, the appropriate components were again added to the reactive bulk composition. This iterative modeling approach was applied to each successive growth zone. The resulting P-T estimates are reported in Figure 9b, with error bars showing the P-T uncertainty related to the analytical error of the garnet composition (Lanari et al., 2017). For any given reactive bulk composition, narrowly spaced isopleths return small uncertainty envelopes, whereas widely spread isopleths return larger ones. A detailed description of these methods is presented in the companion paper (Giuntoli et al., in review).

## 5.3 Results of thermodynamic modelling of garnet growth zones

PT estimates obtained from the garnet growth zones are summarized here with the main purpose of distinguishing between pre-Alpine (HT) and Alpine (HP) growth periods. The full dataset used is provided and discussed in more detail in Giuntoli et al. (in review). Figure 9 summarizes the composition of all the garnet growth zones in a ternary diagram (Alm, Prp, Grs) and the optimal P-T conditions determined for each growth zone. Notably, in each sample studied the crystals analysed in thin section all show the same chemical zoning pattern.

Some preliminary remarks are needed here before presenting the detailed results for each sample. Garnet cores in all modelled samples indicate growth at granulite facies conditions (Fig. 9). In sample FG1249 only the garnet core shows an overgrowth (Rim1) for which modelling indicates amphibolite facies conditions (Fig. 9). Rim1 in this sample is considered to be pre-Alpine, since amphibolite to granulite facies conditions are well established to be upper Paleozoic (Kunz et al., 2017) in the terrane sampled, i.e. the central Sesia Zone. HT conditions at low to intermediate pressures have never been reported for the Alpine metamorphism, which is of eclogite facies grade. Indeed, all other garnet rims in the samples studied indicate eclogite facies conditions, hence are attributed to the Alpine cycle. This is in line with the HP mineral inclusions of phengite, glaucophane and rutile (see section 4; Figs. 5b, 6b, 7b, S5b, S6b) in garnet, as well as our results of P-T modelling (Fig. 9). The sealed inner fractures did not require special consideration in modelling because of their low modal abundance («1 volume %). However, we note that their composition matches that of Alpine rims (e.g. Rim2 in sample FG1315). Conditions found from one sample to the next show minor differences, indicating spatial and/or temporal gradients in the P-T conditions. Given that the terrane sampled contains several tectonic slices (Giuntoli and Engi, 2016), it is certainly possible that these conditions may not have been recorded in all samples at the same time of the evolution of the belt.

FG1315: The garnet core is predicted at ~0.8 GPa and 750 °C. The HP Alpine rims are modelled at ~1.5 GPa and 650°C, 1.9 GPa and 650°C, 1.8 GPa and 670°C respectively.

FG12157: Modelling indicates ~0.6 GPa and 900 °C for the core; Alpine Rim1 and Rim2 are predicted at ~1.6 GPa, 650°C and ~1.4 GPa, 630°C, respectively.

FG1249: The pre-Alpine core is modelled at ~0.6 GPa and 730°C; 10 vol% of garnet core is predicted to have crystallized (Fig. 10). Rim1 grew at pre-Alpine amphibolite facies conditions (~0.6 GPa and 620°C); GrtMod predicted resorption of 2 % of the core and growth of 2 % Rim1. Alpine Rim2 crystallized at ~1.6 GPa, 620°C; the model yields 10 vol% growth of

Rim2 and less than 1 % resorption of previous generations. Rim3 crystallized at ~1.5 GPa, 660°C with 7 % resorption of garnet Rim2 and growth of 9 % Rim3.

FG1347: The pre-Alpine core is modelled stable at ~0.8 GPa and 770°C. All Alpine rims (1-3) are modelled stable between 1.7-2.0 GPa and 580-600°C.

## 6 Discussion

As summarized in the previous section, garnet cores are modelled stable in granulite facies conditions; Rim1 of sample FG1249 suggests amphibolite facies conditions. In situ U-Pb dating of metamorphic growth rims in zircon from all the samples presented in this study gave lower Permian weighted mean ages (~295-280 Ma; results presented in Kunz et al., 2017). These data are in agreement with the late Permian retrograde metamorphism from granulite to amphibolite facies conditions proposed by Lardeaux and Spalla (1991) and Rebay and Spalla (2001). Except for Rim1 in FG1249, all of the garnet rims modelled are found to be stable at eclogite facies conditions that are similar to those recently determined for the central Sesia Zone (e.g. Konrad-Schmolke et al., 2006;Konrad-Schmolke and Halama, 2014;Regis et al., 2014;Rubatto et al., 2011;Lanari et al., 2017). In the following, the main textures recognized in garnet are discussed and related to specific processes.

### 6.1 Micrometre-size fracture network in garnet cores

A network of fine fractures is present in all of the pre-Alpine cores of the samples (inner fractures in Fig. 3c; Fig. 4). These fractures have widths from few microns to some tens of microns and irregular shapes, with sharp edges and conjugate systems with 90° interception. These fractures are not localized around inclusions and do not show a radial distribution, so inclusion-induced fracturing during decompression is ruled out as a mechanism of formation (e.g. Whitney, 1996;Wendt et al., 1993). The fractures are not visible by optical or scanning electron microscopy, as they are sealed by a garnet similar in composition to the first HP Alpine generations. These small-scale fillings are considered coeval with the formation of one of the first Alpine garnet rims, as they are similar in chemical composition.

In outer parts of the core fractures are less abundant, and the more calcic garnet sealed cracks show sharp boundaries against the old garnet core composition. In central parts the core appears "cloudy", owing to the dense network of sealed cracks, so the finely spaced sealed cracks are more difficult to discriminate. Minor diffusional smoothing may have occurred over a scale up to ten microns. In the present samples, the fracture network in garnet cores is best observed in the $X_{Grs}$ maps, but is also evident in other X-ray maps, notably of Fe and Mg, but less so for Mn (except for FG1249), possibly because the concentration is too low to detect such a small variation in Mn. Grossular fraction has long been known to be strongly pressure-dependent (Kretz, 1959), and pressure is the physical variable expected to vary particularly in a subduction setting. Moreover, Ca in garnet is among the divalent elements least affected by diffusion up to fairly high temperature (e.g. Carlson 2006), so it is most likely to retain growth features.

The network of cracks observed in our samples probably reflects brittle deformation of garnet, which implies a small ductility contrast between garnet and the matrix (e.g. Raimbourg et al., 2007). High strain rates that induced these cracks may well have been related to seismic failure (Austrheim and Boundy, 1994;Austrheim et al., 1996;Austrheim et al., 2017;Angiboust et al., 2012;Wang and Ji, 1999;Hertgen et al., 2017). Note that pervasive deformation, e.g. by shearing, would have dismembered the fractured garnet cores, which is not observed. Since the mineral assemblages formed by Permian granulite facies metamorphism were mostly anhydrous, the rheological contrast between garnet and its matrix is expected to have been small, up to the HP Alpine (re)hydration (Engi et al., in review). As we show in the following, fractures in a rheologically strong mineral like garnet and their subsequent sealing thus provide a link from the Permian HT- to the Cretaceous HP-conditions, more specifically related to the first stage of fluid influx (Fig. 10a). Apart from such fractures in the garnet cores, pre-Alpine or other pre-eclogite facies fabrics are but rarely preserved in the study area because Alpine metamorphic re-equilibration at eclogite facies conditions can reach nearly 100% (see also section 6.4). Specifically, the brittle stages visible at micron scale in garnet cores could not be directly linked to any locally or regionally prevalent strain patterns.

Nevertheless, we cannot rule out that some of the fractures developed in pre-Alpine time, notably in an extensional tectonic setting. This has been proposed for other localities (Floess and Baumgartner, 2013), where garnet breakdown occurred at LT and LP, involving a volume increase and hydration. However, in the present case, micrometer-wide fractures dissect pre-Alpine garnet, and the coherence of these fractured grains is preserved. We found no chemical evidence of any LT- or LP-alteration predating the growth of Ca-enriched garnet, except for one sample (FG1249, discussed below). All these observations indicate an Alpine origin for the fracture network and suggest that these were healed by a new garnet generation before the fragments were dismembered by subsequent strain. Most probably the formation of fractures and growth of garnet inside these fractures were closely related, and mineral compositions indicate that this occurred at eclogite facies conditions. Moreover, comparable garnet textures in a similar geological context (Mt. Emilius klippe) have been interpreted to reflect fracturing and sealing at Alpine eclogite facies conditions (Pennacchioni, 1996;Hertgen et al., 2017;Angiboust et al., 2016)

Brittle behavior of garnet is also manifested during the retrograde Alpine greenschist facies conditions (0.15-0.3 GPa and 300-350°C) in other samples from the IC of the Sesia Zone. This deformative stage produced textures varying from fragmented garnet, showing no displacement, to trails and pods of garnet (several hundred microns in size) that represent torn-apart garnet porphyroclasts (Trepmann and Stöckhert, 2002;Küster and Stöckhert, 1999). Such fractures cut through all the garnet growth zones (Fig. 4 in Trepmann and Stöckhert, 2002), these are not healed by garnet, instead chlorite frequently crystallized along them. Fractures in garnet are filled by retrograde metamorphic minerals (e.g. chlorite, feldspar, epidote) if they form out of the stability field of garnet (e.g. Prior, 1993). Based on these criteria, we suggest such a mechanism for the millimeter-long fractures (named late fractures in Fig. 3c, see section 5.1) we found in our samples. These cut through all the garnet generations and are completely different from the fine fractures discussed above.

## 6.2 Resorption and growth: fluid-related textures

Our samples record several stages of resorption with unequal portions of garnet being affected (Fig. 4). Resorption is most extensive in sample FG1315, as judged by textures: Lobate edges, peninsular structures, veins inside Rim1, and formation of atoll garnet (Figs. 4a, 5, 10b, S5). In particular, the vein network dissects Rim1, and resorption is evident along these veins and on both sides of Rim1, prior to growth of Rim2. Note that Rim2 displays the same chemical composition in all of these domains. For this reason, we surmise that Rim1 and 2 are not a simple growth sequence, with the older, internal generation being overgrown more externally by a younger rim. Instead, the geometry indicates that the older growth zone (Rim1) was partially resorbed and replaced by a younger one (Rim2) that precipitated on both sides of Rim1. This interpretation is also supported by numerous micrometre-size rutile inclusions in Rim1; these highlight paleo-porosity, an essential feature of the replacement processes (e.g. Putnis, 2015). Analogous zoning patterns are evident also in samples FG12157 and FG1249 Rim1 and 2, even though the vein network in these samples is less clearly visible in the compositional maps.

Four processes have been proposed to account for the formation of atoll garnet: (a) simultaneous multiple nucleation and coalescence processes (e.g. Cooper, 1972;Spiess et al., 2001), (b) rapid and short-term poikiloblastic growth (e.g. Atherton and Edmunds, 1966;Ushakova and Usova, 1990), (c) changes in the stoichiometry of the garnet-forming reaction (Robyr et al., 2014), and (d) dissolution of the core that was rendered unstable (by changes P-T conditions), and precipitation of new garnet stable in the presence of circulating fluid (e.g. Smellie, 1974;Homam, 2003;Cheng et al., 2007;Faryad et al., 2010;Ortolano et al., 2014a). In sample FG1315 textures in large garnet cores (up to several mm in size) reflect partial resorption, showing lobate structures or peninsular features. Detailed thermodynamic modelling of this sample (Giuntoli et al., in review) predicts extensive resorption of garnet cores during the formation of Rim3. Based on this evidence, we conclude that atoll garnet formation was related to process (d), i.e. partial dissolution of the unstable HT-core under HP-conditions in the presence of a fluid. Note that the effect of this process is grain size-dependent, and atoll garnet cores observed in this sample are indeed limited in size from 50 to a few hundred microns. So, while destabilization of these small garnets lead to complete replacement of the core, the latter was not entirely replaced in the large garnets (Fig. 10b). In FG1315, resorption textures are also visible in the Permian growth zone of zircon, as documented by Giuntoli et al. (in review). Resorption occurred at several stages, as evident from the peninsulas in which garnet Rim1 crystallized and was successively enlarged by Rim2 and by Rim3. Figure 10b summarizes the growth / resorption chronology of garnet in this sample.

Resorption and growth textures in sample FG12157 are similar to those in FG1315, except that no atoll garnet is observed. However, a drastically different type of textural features is found in sample FG1249: Neither clear lobate structures nor peninsulas indicate resorption, except for a narrow Rim2 that grew at the expense of garnet core and Rim1. The growth chronology of the sample FG1249 is summarized in Fig. 10a. In sample FG1347, resorption traces are limited close to the fracture network in the core, with minor resorption of Rim2 (Fig. 4d).

With one exception, all of the samples presented above also contain zircon and sparse monazite; these are only preserved as pre-Alpine (Permian) relics in the core of Alpine allanite. The exception is FG12157 where monazite is completely pseudomorphed by allanite and apatite; (Giuntoli et al., in review). Apart from garnet and these accessory relics, the main pre-Alpine HT assemblage has completely re-equilibrated at eclogite facies conditions. This is also evident from the volatile contents of these samples and their high percentages of hydrous minerals: LOI (loss-on-ignition) values range from 1.58% to 2.44% (Table S1) compared to values of 0.5-0.7 wt% for proposed equivalent rocks in the Ivrea Zone, i.e. pre-Alpine upper amphibolite to granulite facies samples lacking an Alpine facies overprint (Engi et al., in review). As detailed in that study, this implies that the nominally dry pre-Alpine HT assemblages were replaced in response to disperse infiltration of hydrous fluid. This Alpine hydration process occurred in several pulses or stages (Fig. 10). Previous partial hydration is evident only in one sample (FG1249), in which a first rim formed during pre-Alpine retrogression from granulite to amphibolite facies conditions. In all other cases studied, the increase in bulk rock volatile contents happened during the Alpine HP-evolution, when fluid was imported while the IC was in the subduction channel. This hydration is expected to have enhanced ductile deformation and metamorphic reaction rates (Austrheim, 1990;Austrheim et al., 1997;Pennacchioni, 1996).

## 6.3 Re-equilibration close to fluid pathways

As described in section 5.1.3, fractures in the pre-Alpine garnet cores are evident in X-ray maps made from sample FG1249, and all garnet end member maps show that these are sealed by garnet of a composition ($Alm_{62}Prp_{18}Grs_{18}Sps_2$) similar to the two Alpine rims (Rim2 and Rim3). Note that the core is chemically uniform, as expected in granulite facies garnet owing to fast diffusion at high temperature (e.g. Caddick et al., 2010). However, in an area some 500 µm wide (indicated by white arrows in Fig. 7), garnet is affected by fractures, and the $X_{Alm, Prp, Sps}$ maps show that it is compositionally different from the core, except for a few islands where original granulite facies garnet remains, and it displays an Alpine composition (Figs. 7, 8). This feature is best observed in the Mn-map, which serves as a marker delimiting the original pre-Alpine garnet core with its high $X_{Sps}$ values (0.05-0.06; Fig. 7f). Overprinted areas show a sharp decrease in $X_{Sps}$, down to values of 0.02. The $X_{Grs}$ map (Fig. 7c) is drastically different, as garnet displays the same composition as pristine pre-Alpine generations ($X_{Grs}$ 0.05 and 0.09 for core and Rim1, respectively), except just a few microns away from the fractures.

This zoning pattern suggests that fractures provided fluid access, initiating a re-equilibration process that affected the surrounding volume of garnet over a distance of several hundred microns. As minor fractures are visible branching from the major ones, we consider this area as a damaged zone in which re-equilibration of the garnet was enhanced along a network of micro-fractures and in their proximity, due to the increased area/volume ratio (Austrheim et al., 1996;Pennacchioni, 1996). Erambert and Austrheim (1993) reported similar fracture patterns in pre-Caledonian granulite garnet of the Bergen Arcs (Western Norway), and they interpret these as fluid channels along which element mobility was enhanced. These studies concluded that re-equilibration of granulite-facies garnet during Caledonian eclogite conditions occurred proportionally to the fracture density and fluid availability.

In the present samples, re-equilibration processes affecting garnet were evidently much more effective for Alm, Prp, and Sps than for Grs. We propose that diffusion of Ca ions through the damaged lattice of the original garnet was slower than of $Fe^{2+}$, Mg, and Mn. Conspicuously similar phenomena were observed by Erambert and Austrheim (1993). Their evidence is in line with what we discussed above, i.e. that new garnet formed just along fractures; in neighbouring areas strong overprinting of original garnet is observed for Prp, Alm and Sps but not for Grs. Raimbourg et al. (2007) advanced that intra- and inter-granular transport of Ca in garnet is inefficient compared to $Fe^{2+}$ and Mg. Volume diffusion of divalent ions is slow in garnet at temperatures below 700 °C (Ganguly, 2010), but differences in interdiffusion between Ca, Mg, Fe, and Mn are notable (e.g. Carlson, 2006).

In the latest phases of this re-equilibration process, brittle fractures would have been sealed by the Alpine garnet visible along the fracture walls. This may have occurred in several stages, as can be inferred from the $X_{Grs}$ map, where garnet sealing the fractures has the composition of Rim2 and Rim3 (Figs. 7, 8).

$X_{Sps}$ can be used as a tracer of re-equilibration processes also in samples where the effect of such processes on garnet texture it is not as striking as for FG1249. This is the case in samples FG12157 and FG1347, in which high fracture density in the core corresponds with low spessartine values (Figs. 6f, S6f). Furthermore, zoning found in the core of FG1347 for Alm, Prp and Sps is not evident in the $X_{Grs}$ map, again indicating that re-equilibration in the proximity of the main fluid pathways was more effective for $Fe^{2+}$, Mg, and Mn than for Ca ions.

## 6.4 Hydration through the IC

The previous sections used garnet textures linked to the main paragenesis and P-T modelling results from Giuntoli et al. (in review) to show that the IC of the central Sesia Zone in the Aosta Valley underwent pre-Alpine granulite facies metamorphism with local amphibolite facies retrogression, followed by major Alpine hydration occurring at different stages of the eclogite facies evolution. As discussed in the previous sections, garnet textures suggest a decrease in the amount of fluid-garnet interaction and related pervasive resorption features from internal areas (South East - FG1315 and FG12157) to external area (North West - FG1249 and FG1347) of the IC (Fig. 1b). Such a tendency is supported by field observation (Giuntoli and Engi, 2016), as the only mesoscopic boudin of pre-Alpine amphibolites found in the area of the IC studied (Compagnoni, 1977) is located close to sample FG1347. These amphibolites are only partially re-equilibrated at eclogite facies conditions, and they preserve pre-Alpine hornblende and plagioclase (Compagnoni, 1977;Gosso et al., 2010).

Hydration in internal areas (SE parts) of the IC started during early subduction metamorphism, as recorded by prograde lawsonite (Zucali and Spalla, 2011) and continued up to the highest P-T conditions recorded in the present samples. In external (NW) areas of the IC a second generation of lawsonite provides evidence of hydration occurring right after rocks reached their HP climax, when temperature decreased (Pognante, 1989). This stage was related to $H_2O$-rich (low $XCO_2$) fluids possibly along preferential channels, producing metasomatism, as reflected by local occurrences of lawsonite in high modal amounts (up to 60 vol.-%).

In the central Sesia Zone (mainly in an area south of the Aosta Valley, whereas our samples were taken north of it), Konrad-Schmolke et al. (2006) found evidence that micaschists similar to those reported here also were not fully hydrated prior to Alpine subduction. Based on P-T modelling, that study concluded that extensive influx of hydrous fluid would have been required for these rocks to remain water-saturated between 1.2 and 1.8 GPa. This is similar to our results, but Konrad-Schmolke et al. (2011) invoked open-system pervasive fluid flow in the Tallorno Shear Zone and concluded that hydration occurred at retrograde blueschist conditions, at the expense of eclogite-facies felsic and basic rocks. They suggest that the minimum amount of fluid interacting with the wall-rocks was 0.1-0.5 wt%. In our samples, a retrograde hydration is significant in only one sample (FG12157), for which P-T modelling indicates a second fluid influx at 1.4 GPa and 650°C (Rim2). In addition, several samples in the area we studied do show minor amounts of hydration during retrograde greenschist facies, but these effects are local and minor (<5% retrogression in most samples). A strong greenschist overprint (replacing 50-90% of the eclogite parageneses) is limited to the vicinity of the major tectonic contact between the IC and EC (Barmet Shear Zone; Giuntoli and Engi, 2016). By analogy, we surmise that the hydration stage described by Konrad-Schmolke et al. (2011) is a relatively late phenomenon occurring in a confined area during retrograde blueschist conditions, whereas the main hydration of the Internal Complex in the region of Aosta Valley occurred during the prograde P-T trajectory, at eclogite facies conditions, as described in this study. The timing of fluid infiltration is reflected by Alpine allanite and zircon ages (Regis et al., 2014;Rubatto et al., 2011;Giuntoli, 2016;Giuntoli et al., in review) and reflects a heterochronous evolution in several tectonic sheets, with older ages (85-77 Ma) generally in the internal (SE) areas of the IC, and younger ages (72-55 Ma) towards external areas (NW).

While textural and chemical evidence indicates that external fluid repeatedly interacted with initially nearly dry protoliths at HP conditions, we have no tight constraints on how much fluid entered at what spatial or temporal intervals. However, for each growth stage (within any one sample), the composition of garnet produced is uniform in each grain analysed, whereas their local geometry differs to some extent. This allows a spatial estimate of the reaction volume involved: Interaction of hydrous fluid with the reactive part of the assemblage (i.e. the matrix minerals) must have been at the scale of a thin section (centimetres) at least, except that incomplete reaction progress left garnet relics. Apart from garnet and local accessories (zircon, monazite), no mineral relics of the successive replacement reactions have been detected, indeed these rocks appear otherwise fully equilibrated at eclogite facies conditions.

Unlike in the Norwegian Caledonides, the relationships between deformation structures that acted as fluid pathways (e.g. Austrheim, 1987) is not clearly evident in our samples. Eclogitization of pre-Alpine granulites and amphibolites in the IC seems to be more an effect of pervasive but disperse fluid infiltration, not of channelized flow. At the microscale sample FG1249 may be considered an exception, since it shows tracks of more channelized fluid flow preserved in millimetric garnets.

In Western Norway, lithological heterogeneities do not seem to play a major role in the localization of deformation, whereas detailed mapping shows that the Internal Complex of the Sesia Zone consists of a pile of tectonic sheets, a few kilometres in thickness (Giuntoli and Engi, 2016). These sheets show internal lithological heterogeneity, with micaschist, eclogites, minor

para- and orthogneiss interlayered at scales from a few millimetres to about a hundred metres. The rheological contrasts among these different rock bodies might well have been where small-scale shear bands nucleated, which served as conduits for fluid infiltration (Oliver, 1996).

**7 Conclusions**

This study documents complexly zoned garnet in polyorogenic rocks that were extensively deformed and (re)hydrated during Alpine eclogite facies metamorphism. Garnet in pelitic schists commonly preserves a porphyroclastic core that reflects medium-pressure HT-metamorphism. These granulite facies conditions prevailed during the Permian and converted the sedimentary protoliths to mostly anhydrous assemblages, which are rarely preserved in the area studied. In the sample suite reported here, pre-Alpine conditions are evident just in garnet cores and relic zircon (Kunz et al., 2017). The relic features show brittle deformation textures, i.e. cracks, but no displacement. These textures reflect high strain rates at the onset of Alpine tectonics, and they may have been generated by seismic failure. In garnet, a network of micrometre-size fractures crosscut the core (and locally a first Alpine overgrowth), and these cracks were sealed by new generations of garnet. Their composition is more calcic and corresponds to the Alpine HP-rims of garnet.

The interaction of percolating fluid with garnet repeatedly produced resorption features, such as lobate structures, peninsulas, or atoll garnet. Textures indicating limited resorption are less common. In rocks where these features are present, re-equilibration produced by intracrystalline diffusion is located in close proximity to brittle fractures, which acted as fluid pathways. Re-equilibration of Mg, Fe and Mn due to diffusion enhanced by fluid occurred over distances of several hundred micrometres, while Ca re-equilibrated over a much smaller distance (few micrometres), indicating slower diffusion. The intensity of prograde hydration and HP-overprinting, as reflected in the preserved garnet textures, varies among the samples analysed. The set as a whole indicates regional differences, as notably the fluid-garnet interaction intensity appears to a decrease (from SE to NW) over a few kilometres within the Internal Complex of the Sesia Zone. In some areas of the same complex, other studies found another phase of decompression-related (blueschist facies) hydration that appears to be related to a major shear zone. In most external (NW) parts of the Sesia zone studied here, a later, strong greenschist overprint is dominant.

This study shows that compositional (X-ray) maps linked with petrographic and microstructural analysis serve as powerful tools to document detailed mineral textures, particularly in garnet. When combined with thermodynamic modelling results, such images facilitate the analysis and permit quantification of local processes that act at (sub)grain scale, such as local growth and resorption, which play a critical role in hydrating HT protoliths and converting them to eclogite facies assemblages.

**Data availability**

Original data underlying the material presented care available by contacting the authors.

**Competing interests.**

The authors declare that they have no conflict of interest.

5    **Acknowledgements**

We thank Martin Robyr for his help in performing EMPA. Discussions with him and Jörg Hermann have been particularly fruitful. We acknowledge constructive comments and suggestions from Gaetano Ortolano and an anonymous referee. Patrice Rey is warmly thanked for editorial handling. This work was supported by the Swiss National Science Foundation (Project 200020-146175).

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

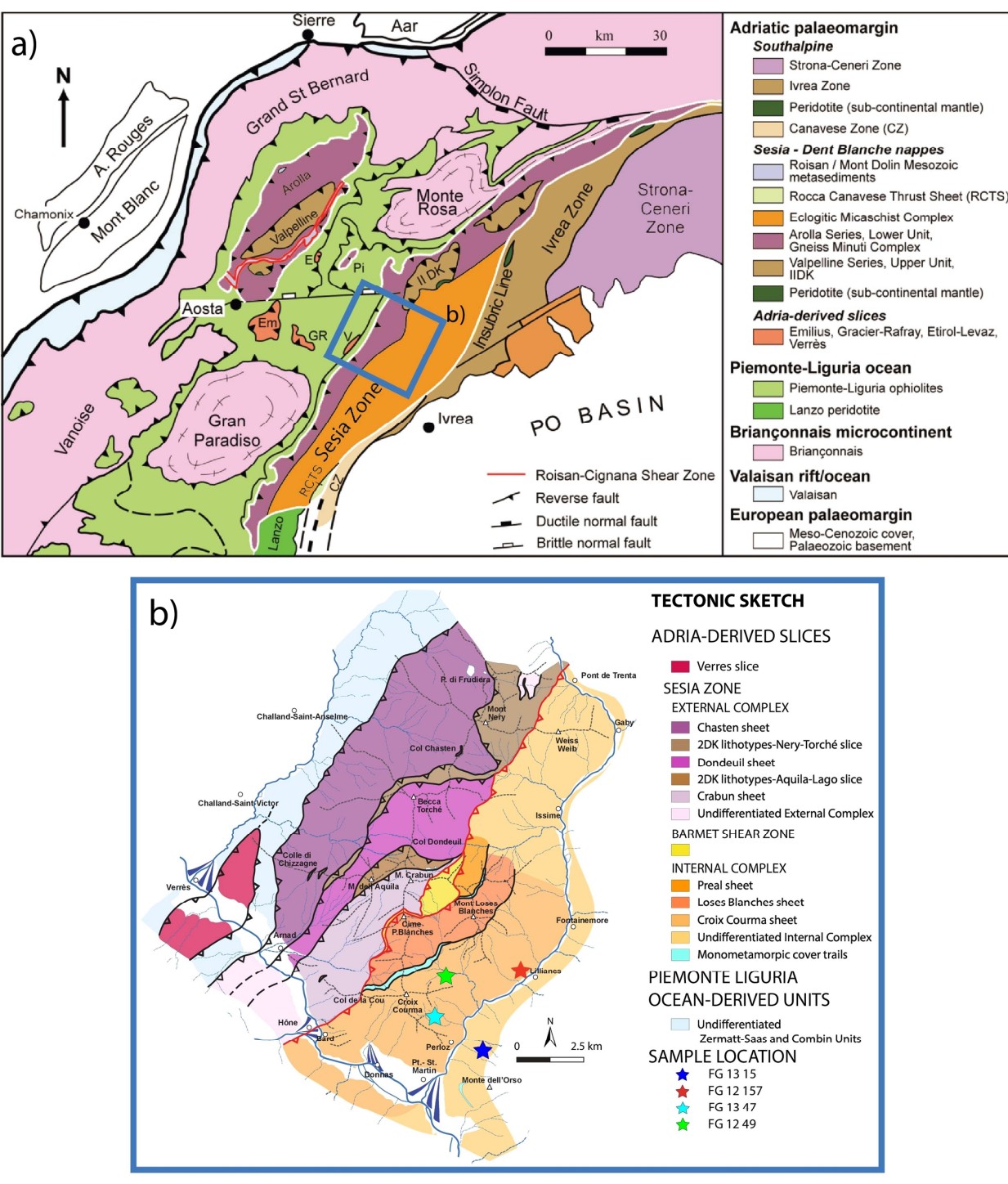

**Figure 1: (a) Simplified tectonic map of the Western Alps (modified from Manzotti et al. (2014), location of study area. (b) Tectonic sketch of study area with sample locations (modified from Giuntoli and Engi, 2016).**

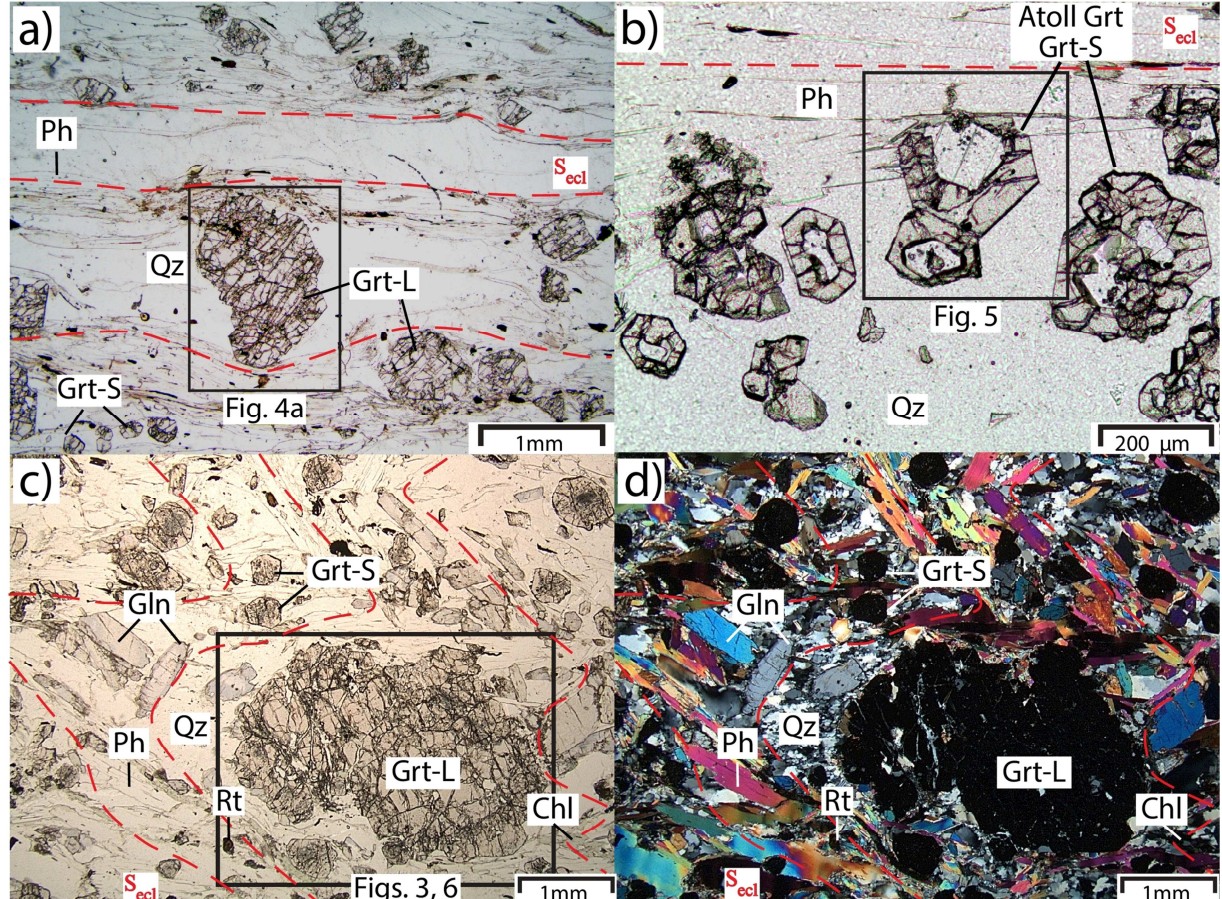

**Figure 2: Thin section optical microphotos displaying the eclogitic foliation ($S_{ecl}$, red dashed line) and the bimodal size range of garnet (large garnet grains Grt-L; small garnet grains Grt-S; see text for further details). Black squares indicate the location of the high-resolution X-ray maps presented in the following figures. (a) $S_{ecl}$ wrapping Grt-L grains in sample FG1315. (b) Same thin section as (a) with atoll garnets (Grt-S) located within a Qz rich band and a Ph rich band. (c) and (d) Folded $S_{ecl}$ marked by Ph, Gln and Rt; Chl is present in the fold hinges. Plane-polarized light: (a, b, c); cross-polarized light (d).**

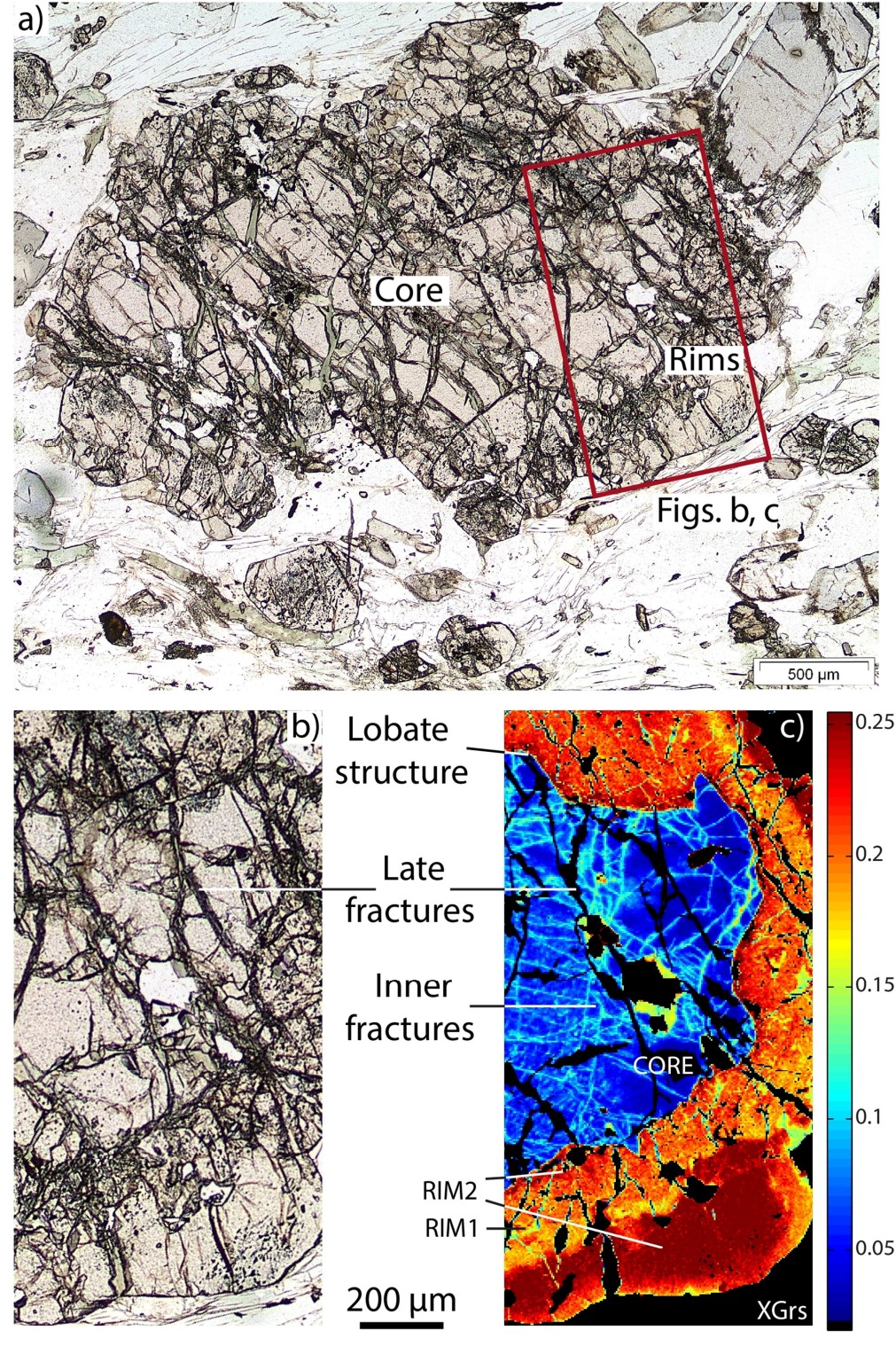

**Figure 3: Garnet porphyrocryst in FG12157, (a) and (b): Optical microphoto, plane polarized light. (a): Clear lobate core, dark inner rim, speckled with inclusions (mostly rutile, up to 20 μm long), outer rim more clear with only minor inclusions. (b) Enlargement shows later fractures (dark) that dissect the entire garnet and contain chlorite. c Grossular X-ray map; blue Ca-poor core reveals numerous micrometre-size fractures sealed by more calcic garnet; sparse late fractures (black) in core and rims. Compare to complete maps in Figs. 4b, 6.**

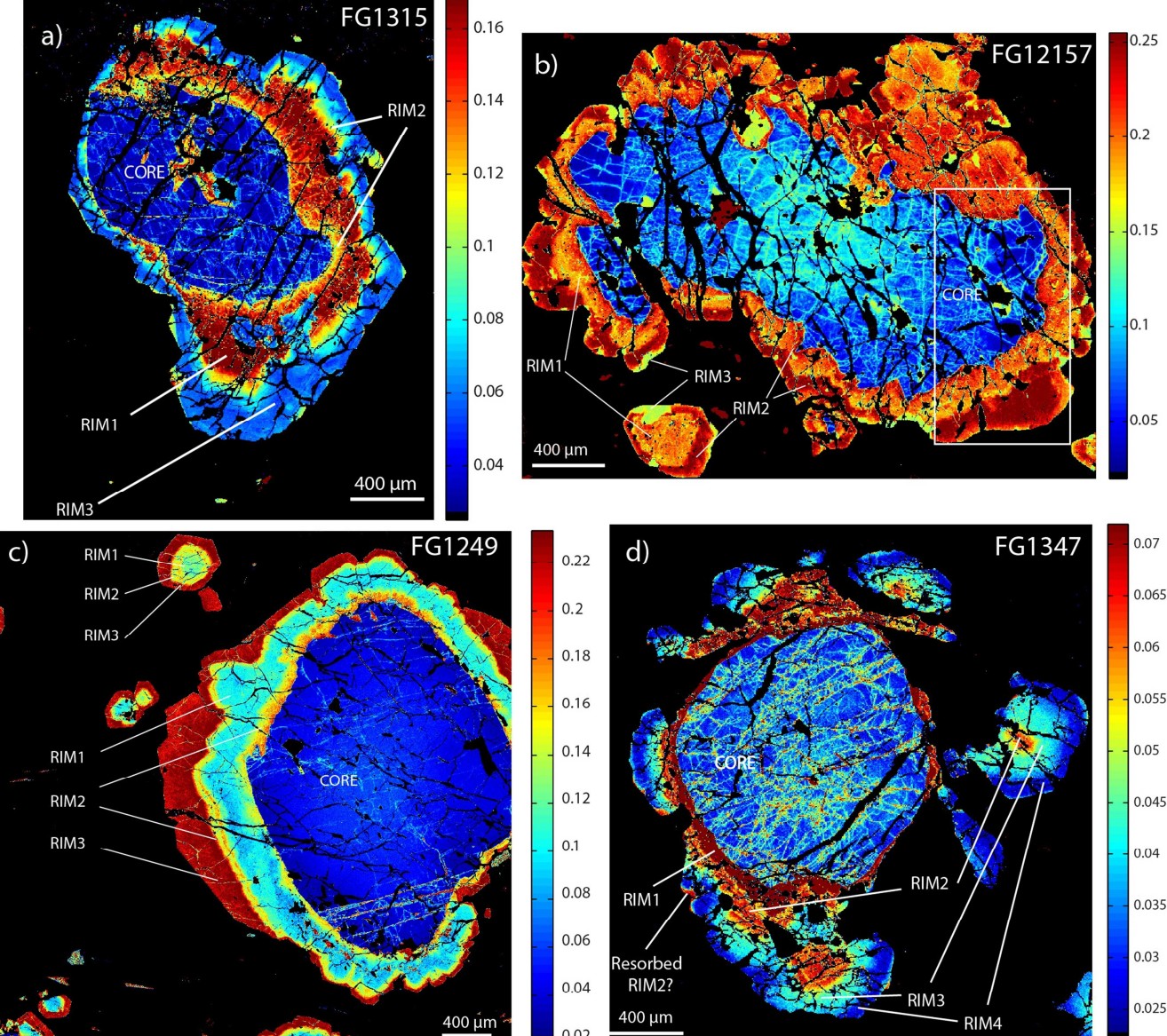

**Figure 4: Growth zones visible in standardized X-ray maps for the grossular end-member. The foliation in all photos is horizontal. Note that a fractured core poor in $X_{Grs}$ is present in all samples and always rimmed by several garnet growth zones. In FG1315 (a) and FG12157 (b) textures indicating resorption include lobate edges and peninsulas (the white rectangle indicates the location of Figs. 3b, c). In FG1249 (c) resorption is inferred from the growth of Rim2 both inside and outside Rim1. In FG1347 (d) resorption is** **limited to the fractures in the core and Rim2 (see text);** **rims vary in thickness in each generation of overgrowths.**

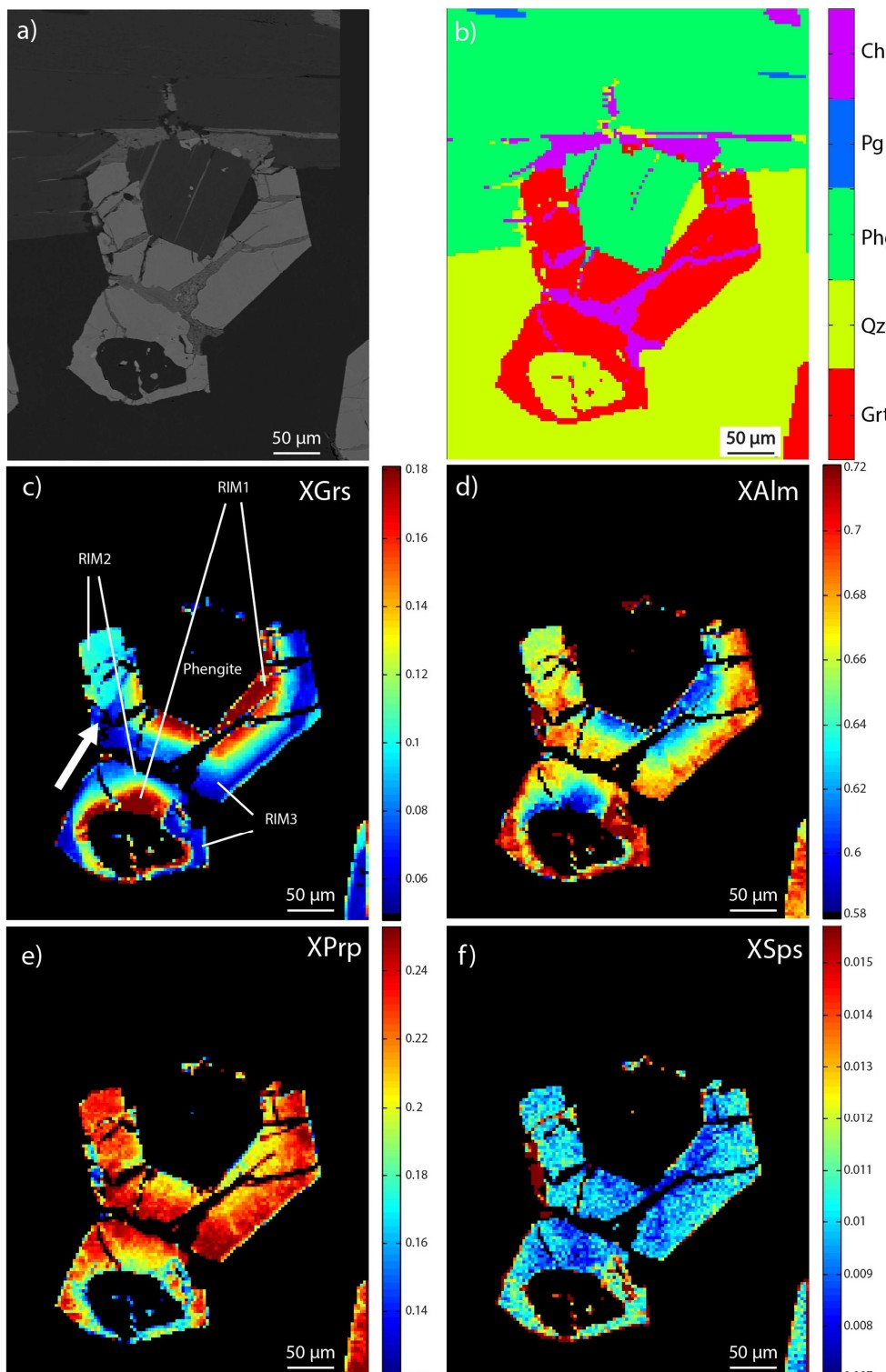

**Figure 5: Small atoll garnet (ø ~100 μm) in sample FG1315. (a) BSE picture. (b) Mineral phases (determined from X-ray maps). Note inclusions of quartz and phengite in the atoll core; fractures dissecting the entire garnet are filled by chlorite. Garnet occurs at the contact of quartz- and phengite-rich bands that define the main foliation. (c) Standardized X-ray map for the $X_{Grs}$ end-member. Note the analogous zoning pattern as for large garnet (Fig. S5), with same $X_{Grs}$ contents and a Rim3 peninsula extending into the resorbed rims (arrow). (d), (e), (f) Standardized X ray maps for $X_{Alm}$, $X_{Prp}$, and $X_{Sps}$ respectively.**

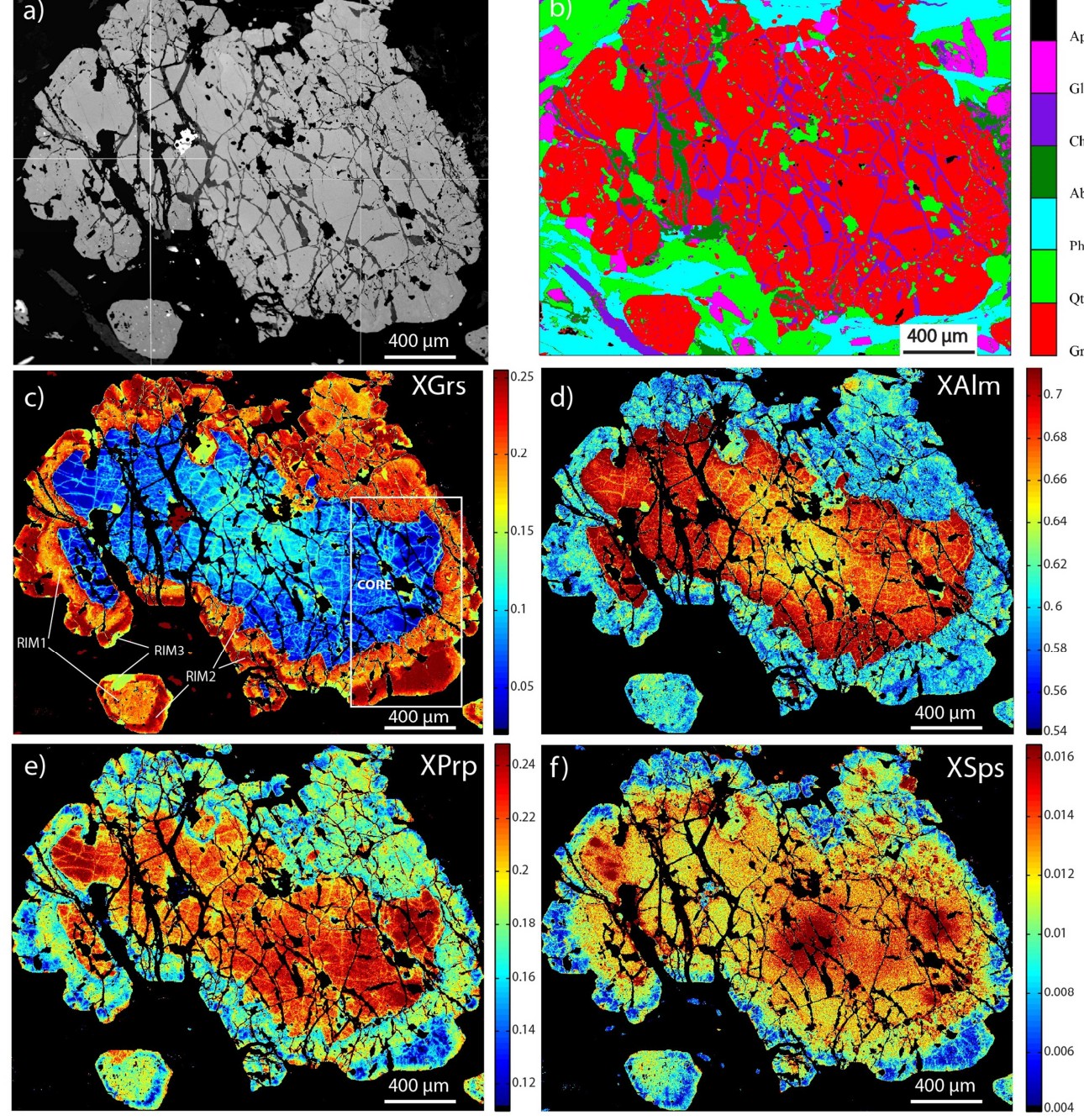

**Figure 6:** Large garnet (ø ~2-3 mm) in sample FG12157. (a) BSE picture with a bright porphyroclastic core and a darker rim. (b) Mineral phases (based on X-ray maps). Quartz inclusions are present in several garnet growth zones, but locally mark the core-Rim1 boundary; note that fractures (filled by chlorite and albite) dissect the entire garnet

grain. Glaucophane and phengite seem to be intergrown with garnet Rim2. **(c)** Standardized X-ray map for $X_{Grs}$ end-member (the white rectangle indicates the location of Figs. **3**b, c. Note the lobate edges along the core and the fracture network within the core that is sealed by garnet higher in $X_{Grs}$. Zoning patterns in small garnet crystals are similar. **(d), (e)** Standardized X-ray maps for $X_{Alm}$ and $X_{Prp}$ display zoning inside the core. **(f)** Standardized X-ray map for $X_{Sps}$. The image appears fuzzy because Mn-contents are low. Note the areas with higher $X_{Sps}$ in the core.

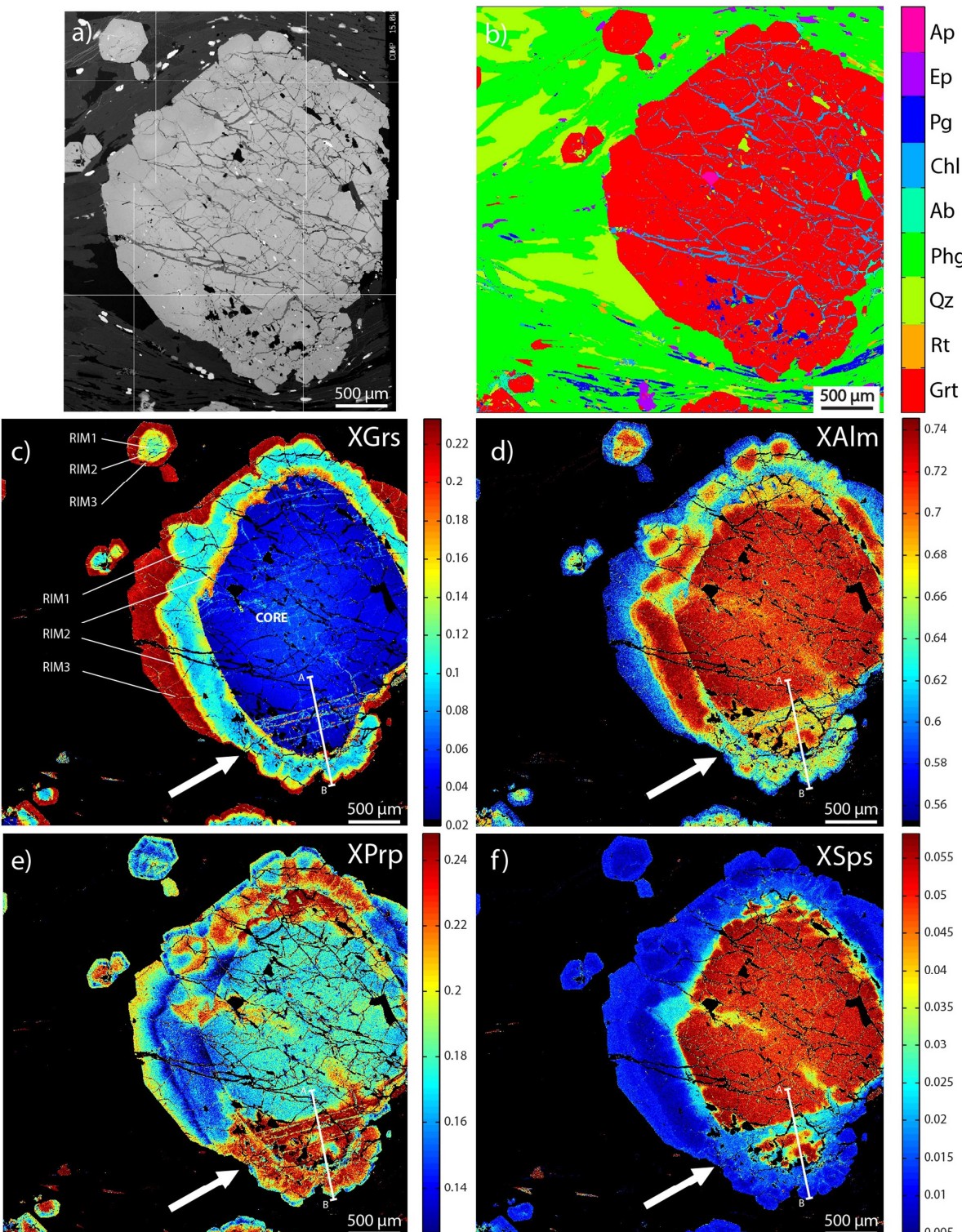

**Figure 7: Large garnet (ø ~3 mm) in sample FG1249. (a) BSE picture shows bright porphyroclastic core and darker rim. (b) Mineral phases (based on X-ray maps). Quartz is included inside the core and at the core-rim boundary, paragonite inclusions are abundant in correspondence of the major fractures sealed by higher grossular garnet. Late fractures that dissect all the garnet are filled by chlorite and albite. (c) $X_{Grs}$ map. Major fractures are sealed by higher grossular garnet. Smaller garnet grains have no core but rims showing the same zoning. (d), (e) and (f). $X_{Alm}$, $X_{Prp}$ and $X_{Sps}$ maps, respectively, suggesting incipient re-equilibration in both core and Rim1 near major fractures (arrows) and along the core-Rim1 boundary (see text for discussion). Relics of the pristine core are preserved as islands inside re-equilibrated garnet Rim 2 and Rim3 (bottom part of figure). Note location of AB profile shown in Fig. 8.**

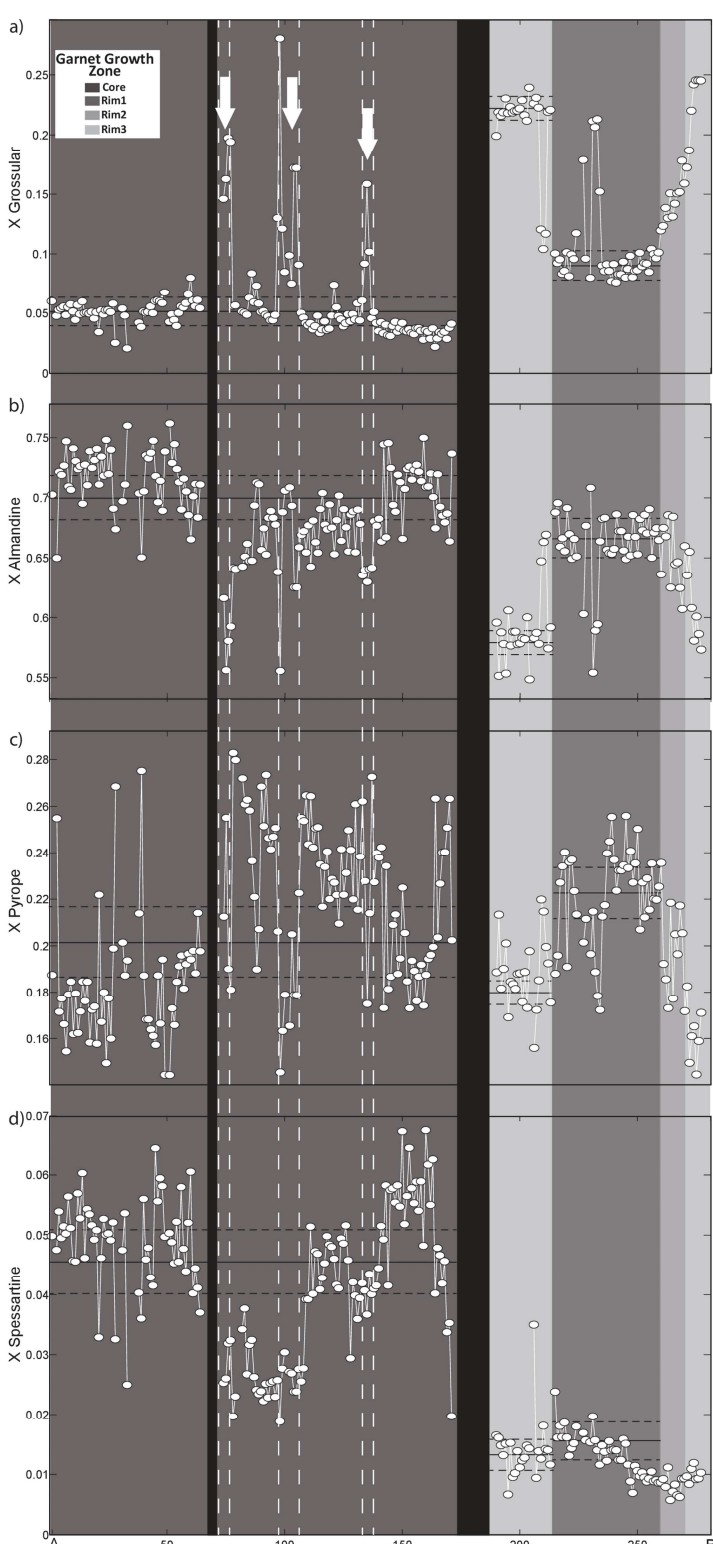

**Figure 8:** Garnet end-member proportions along 300 μm profile (trace shown in Fig. 7) in sample FG1249. Note different mole fraction scales. Late chlorite fractures are highlighted by black bands. The fractures in the core, sealed by a garnet with different composition, are indicated by white arrows. Mean compositions are represented by black solid lines with the standard deviation (1σ) represented by the dashed black lines. (a) Grossular. (b) Almandine. (c) Pyrope. (d) Spessartine. The distinction of each garnet growth zone is based on its grossular content with the aim of helping the comparison.

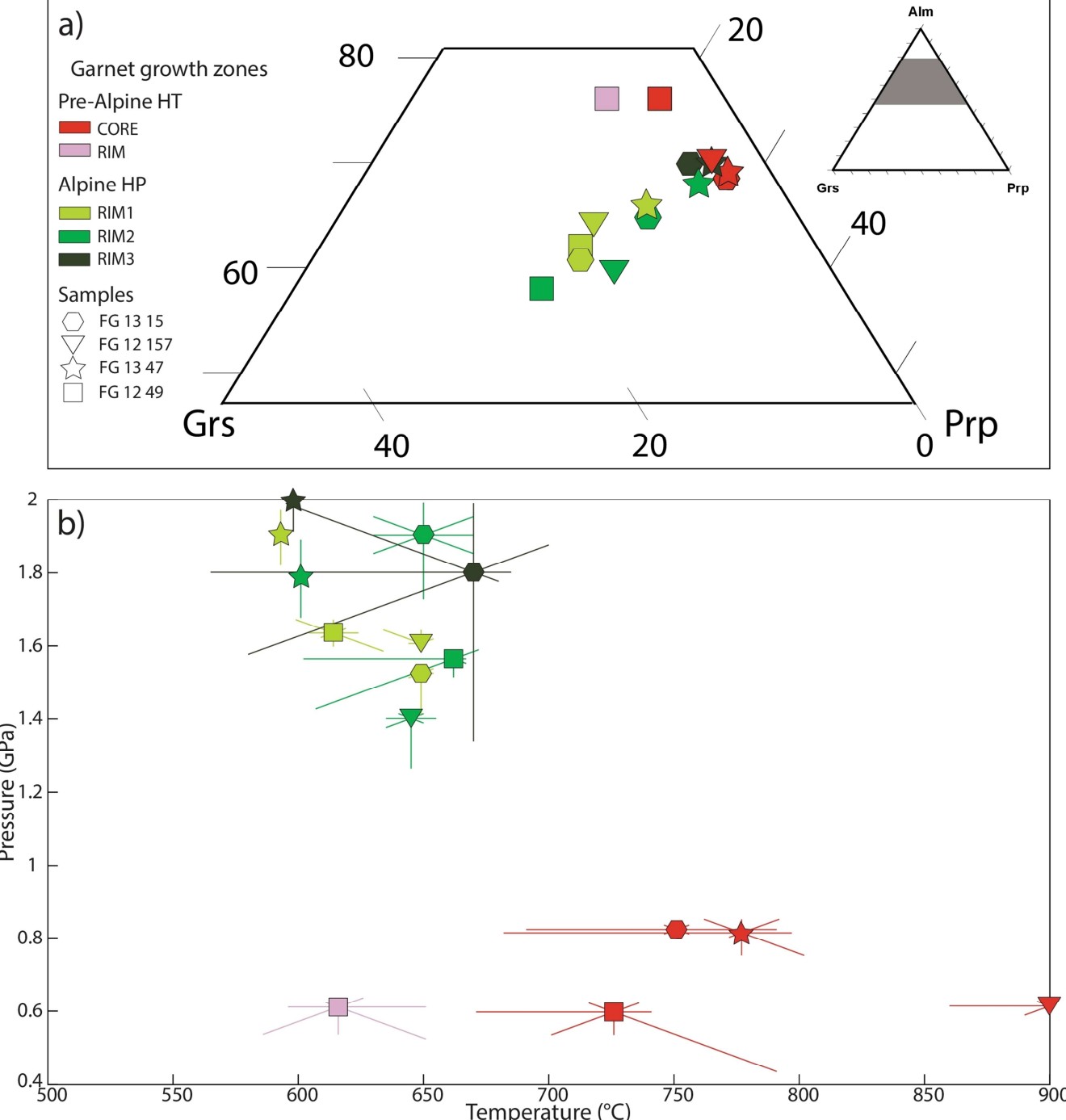

**Figure 9: (a) End member diagram of the garnet growth zones from the four micaschists. (b) P-T diagram with a summary of the modelled conditions for each garnet growth zone. Note major difference in P-T conditions from the pre-Alpine to the Alpine generations. The effect of uncertainties in each set of isopleths is shown around the P-T**

model of each growth zone (dataset from Lanari et al., 2017 and Giuntoli et al., in review) For sample FG1249 the pre-Alpine rim (pink) corresponds to Rim1, Rim 1 to Rim 2 and Rim 2 to Rim 3 in the text.

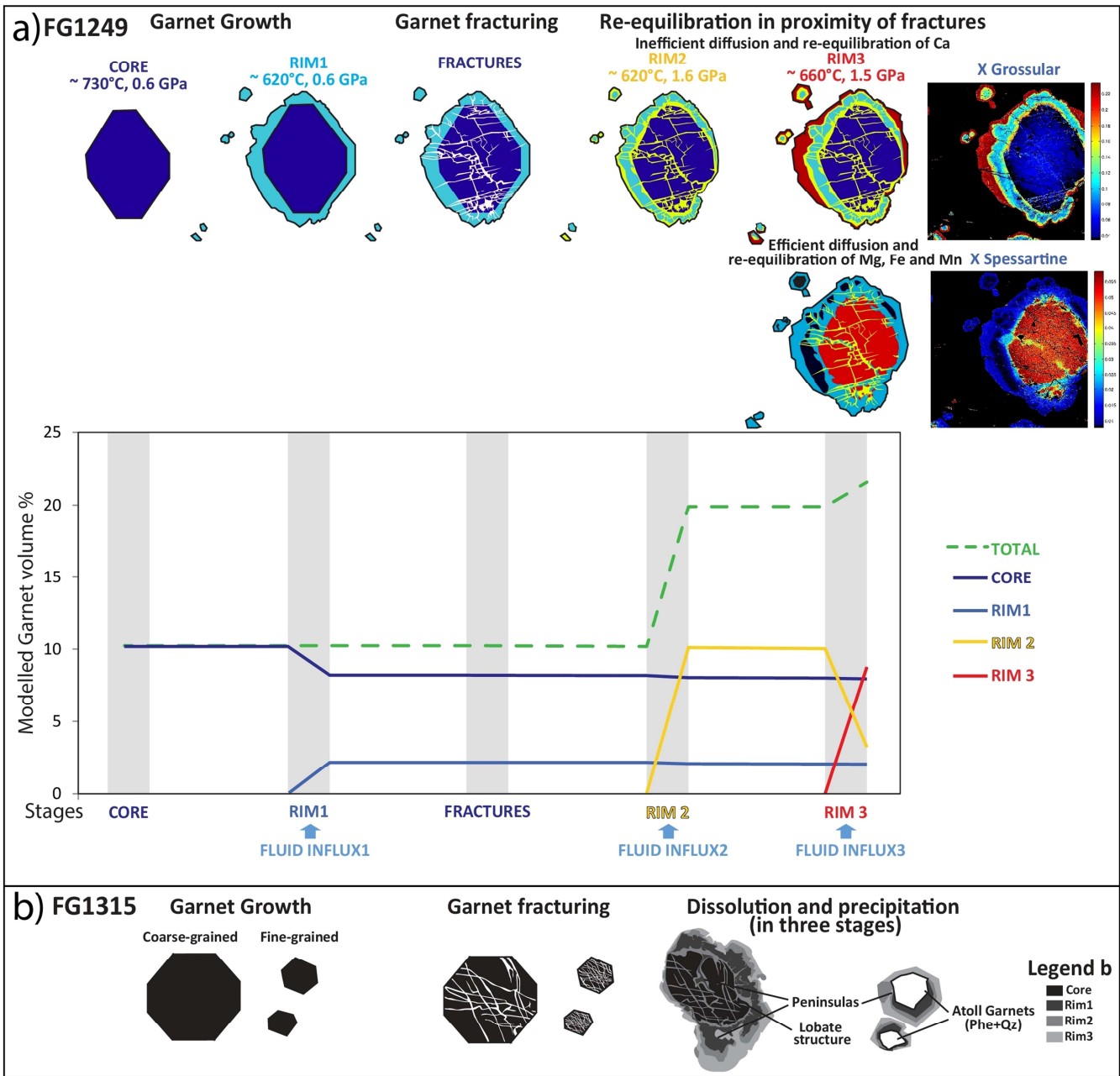

**Figure 10: (a) Sketch illustrating the inferred chronology of sample FG1249, from textural observations and GrtMod thermodynamic modelling (see text for discussion). The preserved succession of stages in garnet evolution is shown along the abscissa, the modelled garnet volume along the ordinate; values include the sum total due to growth and resorption. Note the difference in P-T conditions for the core and Rim1 (both pre-Alpine), relative to Rim2 and 3 (both Alpine); the three fluid pulses are related to Rim1, 2 and 3 growths. After the growth of core and Rim1, a fracture network developed. In proximity of these fractures garnet core and Rim1 got effectively re-equilibrated due to diffusion of Mg, Fe and Mn, whereas Ca remained more immobile. (b) Sketch illustrating the sequence of processes inferred from garnet textures in samples FG1315. Fracture networks are visible in all the garnet core grain sizes. Subsequent stages of fluid influxes caused dissolution of the previous garnet generation/s and precipitation of new garnet, producing peninsulas, lobate structures and atoll structures in smaller garnet grains.**

| Sample | Minerals | Garnet growth zones | Sample locations |
|---|---|---|---|
| **FG1315** | Qz, Ph, Pg, Grt, Ep, Chl, Ab, Rt, Gr, Zrn | Pre-Alpine core + 3 Alpine rims. Note: atoll garnet | Rechantier, X= 408514, Y= 5051580 |
| **FG12157** | Qz, Ph, Grt, Gln, Ep, Chl, Ab, Rt, Zrn, Ilm, Gr | Pre-Alpine core + 2 Alpine rims | Lillianes, X= 409683, Y= 5054033 |
| **FG1249** | Qz, Ph, Pg, Grt, Ep, Chl, Ab, Rt, Gln, Zrn | Pre-Alpine core and Rim1 + 2 Alpine rims | Faye, X= 406637, Y= 5053931 |
| **FG1347** | Qz, Ph, Pg, Cld, Grt, Ep, Chl, Rt, Zrn | Pre-Alpine core + 4 Alpine rims | Liévanere, X= 406318, Y= 5052474 |

**Table1: Micaschists analysed (all from the Lys Valley). Mineral abbreviations from Whitney and Evans (2010). Coordinates refer to ED 1950 UTM Zone 32N.**

| Sample | FG1315 | | | | FG12157 | | | FG1249 | | | | FG1347 | | | | |
| --- | --- | --- | --- | --- | --- | --- | --- | --- | --- | --- | --- | --- | --- | --- | --- | --- |
| Garnet | CORE | RIM 1 | RIM 2 | RIM 3 | CORE | RIM 1 | RIM 2 | CORE | RIM 1 | RIM 2 | RIM 3 | CORE | RIM 1 | RIM 2 | RIM 3 | RIM 4 |
| **Average composition (wt%)** | | | | | | | | | | | | | | | | |
| SiO2 | 37.70 | 37.96 | 37.81 | 37.44 | 38.17 | 38.43 | 38.74 | 36.91 | 36.62 | 37.80 | 37.98 | 37.75 | 37.88 | 38.18 | 37.75 | 37.33 |
| TiO2 | 0.01 | 0.01 | 0.01 | 0.01 | 0.07 | 0.07 | 0.17 | 0.05 | 0.16 | 0.27 | 0.04 | 0.02 | 0.02 | 0.02 | 0.03 | 0.03 |
| Al2O3 | 21.27 | 21.44 | 21.46 | 21.32 | 21.07 | 21.20 | 21.24 | 20.83 | 21.36 | 21.18 | 21.77 | 21.25 | 21.28 | 21.09 | 21.21 | 21.33 |
| FeO | 31.46 | 27.94 | 30.03 | 32.07 | 31.23 | 29.05 | 26.83 | 33.06 | 34.23 | 28.79 | 27.12 | 31.30 | 30.19 | 30.82 | 31.96 | 32.07 |
| MnO | 0.61 | 0.41 | 0.42 | 0.31 | 1.00 | 0.57 | 0.56 | 2.31 | 0.37 | 0.66 | 0.56 | 0.92 | 0.37 | 0.39 | 0.35 | 0.32 |
| MgO | 7.19 | 5.33 | 6.18 | 6.30 | 6.41 | 5.08 | 4.22 | 4.65 | 3.86 | 5.18 | 4.86 | 7.05 | 5.76 | 6.60 | 6.62 | 7.30 |
| CaO | 1.31 | 6.72 | 4.17 | 2.08 | 1.38 | 5.59 | 8.39 | 1.77 | 3.27 | 6.35 | 8.38 | 1.14 | 3.96 | 2.13 | 1.49 | 0.96 |
| Cr2O3 | 0.00 | 0.00 | 0.00 | 0.00 | 0.04 | 0.04 | 0.04 | 0.03 | 0.04 | 0.03 | 0.03 | 0.00 | 0.00 | 0.00 | 0.00 | 0.00 |
| Total | 99.56 | 99.81 | 100.08 | 99.52 | 99.37 | 100.03 | 100.18 | 99.62 | 99.90 | 100.27 | 100.73 | 99.41 | 99.47 | 99.24 | 99.41 | 99.33 |
| **Formulae based on 12 oxygens** | | | | | | | | | | | | | | | | |
| Si | 2.965 | 2.974 | 2.958 | 2.957 | 3.021 | 3.019 | 3.035 | 2.954 | 2.926 | 2.960 | 2.948 | 2.976 | 2.990 | 3.016 | 2.982 | 2.943 |
| Ti | 0.001 | 0.001 | 0.001 | 0.001 | 0.004 | 0.004 | 0.010 | 0.003 | 0.010 | 0.016 | 0.002 | 0.001 | 0.001 | 0.001 | 0.002 | 0.002 |
| Al | 1.972 | 1.980 | 1.979 | 1.985 | 1.965 | 1.963 | 1.961 | 1.965 | 2.011 | 1.955 | 1.991 | 1.974 | 1.979 | 1.964 | 1.975 | 1.981 |
| Fe | 2.069 | 1.831 | 1.965 | 2.119 | 2.067 | 1.908 | 1.758 | 2.213 | 2.287 | 1.885 | 1.760 | 2.064 | 1.992 | 2.035 | 2.111 | 2.114 |
| Mn | 0.041 | 0.027 | 0.028 | 0.021 | 0.067 | 0.038 | 0.037 | 0.157 | 0.025 | 0.044 | 0.037 | 0.061 | 0.025 | 0.026 | 0.023 | 0.022 |
| Mg | 0.843 | 0.623 | 0.720 | 0.741 | 0.756 | 0.595 | 0.493 | 0.555 | 0.459 | 0.605 | 0.563 | 0.828 | 0.677 | 0.777 | 0.780 | 0.858 |
| Ca | 0.110 | 0.564 | 0.350 | 0.176 | 0.117 | 0.470 | 0.704 | 0.152 | 0.280 | 0.533 | 0.697 | 0.096 | 0.335 | 0.180 | 0.126 | 0.081 |
| ∑ cations | 8.001 | 8.000 | 8.000 | 8.000 | 7.997 | 7.998 | 7.997 | 7.998 | 7.998 | 7.998 | 7.998 | 8.000 | 7.999 | 7.999 | 7.999 | 8.000 |
| **Molecular proportions of garnet end members** | | | | | | | | | | | | | | | | |
| Alm | 0.676 | 0.601 | 0.641 | 0.693 | 0.687 | 0.634 | 0.588 | 0.719 | 0.750 | 0.615 | 0.576 | 0.677 | 0.658 | 0.674 | 0.694 | 0.688 |
| Prp | 0.275 | 0.204 | 0.236 | 0.242 | 0.252 | 0.198 | 0.165 | 0.180 | 0.150 | 0.197 | 0.184 | 0.272 | 0.224 | 0.257 | 0.256 | 0.279 |
| Grs | 0.036 | 0.186 | 0.114 | 0.058 | 0.039 | 0.156 | 0.235 | 0.050 | 0.092 | 0.174 | 0.228 | 0.031 | 0.111 | 0.060 | 0.042 | 0.026 |
| Sps | 0.013 | 0.009 | 0.009 | 0.007 | 0.022 | 0.013 | 0.012 | 0.051 | 0.008 | 0.014 | 0.012 | 0.020 | 0.008 | 0.009 | 0.008 | 0.007 |

**Table2: Representative analyses of garnet growth zones**