# Peer review of "Deeply subducted continental fragments: I. Fracturing, dissolution-precipitation and diffusion processes recorded by garnet textures of the central Sesia Zone (Western Italian Alps)"

_Solid Earth, 2017_

## Referee Comment (RC1) · Anonymous Referee #1 · 29 Sep 2017

Summary statement: The manuscript submitted by Giuntoli et al. attempts to contextualize the formation of garnet microstructures and chemical zonation in relation to subduction processes. The manuscript aims to unravel the polymetamorphic and deformation history of the metasedimentary eclogites from the Western Alps. The work presents quality analytical data on garnet zonation and attempts to explain the processes that lead to their formation. Its premise is a beneficial contribution to understanding microscale chemical processes and the dataset is extremely nice. Unfortunately, the data is presented without sufficient context to allow the reader appropriate information to assess the processes proposed. This renders the main conclusions regarding multiple Alpine metamorphic events unjustified, or extremely difficult to support based on the uncertainties within the data set. The manuscript requires the presentation of the complete methods utilized, together with greater contextualization of the behaviour of other minerals within the rock to appropriately understand garnet and therefore eclogite evolution. I understand that some of this information is presented in second installment or in different papers, if the data is reutilized it is not appropriate and leaves this manuscript with no novel quantitative information. This needs to cleared up and explicated stated, as current it is confusing and ambiguous. If this the case the paper is unacceptable for publication in its current form.

Specific comments: The main conclusions ascribing two periods of garnet growth/resorption during the high-pressure Alpine metamorphism is largely unconvincing. The P estimates established for rims2 and 3 are extremely similar, and within uncertainty estimates shown in Figure 8b. It is unknown how these estimates where determined outside of using a program GRTMOD making them hard to believe. What thermobarometry did the program use? What uncertainties are there on the method? What equilibrium volumes were assessed? This information together with the presentation and discussion of the complete thermodynamic models used, and their parameters is needed to justify the estimates provided. The presentation of pseudosections would also provide greater context to the history of the rocks. In particular, the questions surrounding what reactions are controlling garnet growth/resorption? what is the shape and habit of its isopleths? what part of the rock volume was equilibrated? These become difficult to assess in its current form, particularly by the assertion of fluid ingress. The influence on fluid needs to be discussed more fully in terms of the modelling volumes. If the equilibrium volumes are dependent on the diffusion of certain elements additional modelling of chemical potential gradients would be of potential benefit.

In terms of the textural evolution of garnet, greater context of the surrounding phases is required to completely assess the rocks evolution, from both a microstructural and chemical perspective. The sampling is from diverse compositions and protoliths, but consideration of the similarity of the processes is not adequately provided. It begs the explanation of how the period of brittle fracturing is linked to any remnant of current fabrics and their prevailing strain distribution locally and regionally? This manuscript largely states the processes of dissolution and precipitation without providing sufficient evidence to support and disprove possibilities. Presentation of quantitative microstructural information would greatly improve the submission and support some of the conclusions. Much of the focus is on garnet chemistry and not the rocks microstructure.

These issues require contextualization and transparency to support and convince the reader of the interpretations. Many sentences are unclear or have grammatical errors that make it difficult to be sure what the author are trying to convey.

Technical corrections: Page 1 Line 20: "...Sesia Zone, with a general decrease in fluid-garnet interaction observed towards the external areas"

Line 22: 100% of what?

Line 23: what deformation structures? need to look at other phase than just garnet, and even then, you have only talked about brittle fractures. What about crystal plasticity?

Page 2 Line 2: may alter the zonation recorded by garnet

Line 3: diffusion is also dependent on the medium or deformation i.e. grain boundary fluid or pipe diffusion

Paragraph 2: the introduction needs a clearer statement about what the manuscript is trying to solve

Line 18: Permian high-temperature and Alpine high-pressure events. How have these events been determined in age?

[Figure]

Line 19: minerals shouldn't be pluralized

Line 20: clumsy sentenced would benefit from rewording

Line 32: this is an interpretation and needs to be stated as such

Line 33: jumps to generalities of garnet growth from specifics of Alpine rocks, needs to be clearer and explicit following one logic flow then the next

Page 3 Second paragraph: at this stage, it is not clear how you propose to evaluate these possibilities, and exclude different scenarios that lead to your preferred interpretation

Line 9 and 10: confusing sentence needs rewording. As it reads it seems you collected the samples away from the foliation?

Line 11: delete single

Line 12: is the quantitative data already published? This is a major flaw and is not appropriate.

Line 23: delete but

Page 4 Line 3: overprints Line 4: granulite facies conditions and retrograde amphibolite facies conditions

Line 13: latter? Be explicit

Line 15 replace were with involved

Line 22: present the end member formulae used throughout

Line 24: . . .representative amounts of the samples

Line 30: quantify size range

Page 5 I think the use of rim doesn't really apply, they are interpreted growth zones. Using rim makes distinguishing the processes and stages difficult, as in most instance

they are not physically rims. The microstructures, need more detailed descriptions and not left in the supplement. Grains sizes should be quantified. Requires photomicrographs

Line 2: are these exsolution needles? If so this could be of significance to temperature or pressure estimates

Line 6: from the internal and external areas

Line 23: sentence needs rewording

Line 30: I am not sure this is completely true, EBSD provides quantitative information about garnet microstructures, X-ray maps provide links between chemistry and textures. Deleting the word most would help this sentence.

Page 6 Line 1: change pictures to images

Line 10: How was the sealed fractures accounted for in the fractionated compositions and the modelling procedures?

Line 16: how is this accounted for? Growing in two directions

Line 19: providing some quantification of the growth scale in particular directions would strengthen conclusions, particularly across the diverse samples. However this does not explicitly consider crystallography

Line 24: Almandine and spessartine vary a lot in the cores?

Page 7 Line 2: the sealing of the fracture pattern is not obvious on the element maps?

Line 10: again, not obvious what this is talking about; the pyrope contents are unevenly distributed why is this so?

Line 17: remove interpretative statements in the results section

Paragraph 3 No results for the thermodynamic modelling are presented, this is more so thermobarometry. The models should be shown and the methods used to estimate

P-T discussed. What parts of the garnet were fractionated (supplement figure maybe), how did this vary throughout the rest of the rock volume, garnet does not grow in isolation. No equilibration context is provided, it would be good to know the reactions controlling the growth/resorption. What are the nature of the garnet isopleths? Was fluid added and how did it change the modelling? These questions are not explicitly addressed. Discussion of the uncertainties of the P-T estimates are fundamental to resolving these discrete events, currently they are within errors presented in Figure 8 and cannot be resolved. The way this reads it seems the data has been presented elsewhere and therefore is largely re-interpreted and not appropriate as central conclusion of this manuscript, what is new here?

Page 8 Line 2: don't start a sentence with and

Line 7-14: there is an extremely large variation in pressure across all samples, encompassing the entire range of the high-pressure event, how can these be resolved as discrete metamorphic episodes?

Line 17: there is no U-Pb ages presented?

Line 30: replace fillings with more indicative terminology

Page 9 Line 1: veins? Keep terminology consistent

Line 4: best observed, delete visible. But less so for Mn

Line 5: if concentration limits are near the lower limits of detection this should be discussed in the results

Line 6: yes, grossular is commonly pressure dependent, but this should be show specifically for these samples together with the nature of garnet modes to assess potential resorption.

Line 8: sentence needs restructure as it is confusing in its current form

Line 9: these options should be presented as possibilities unless direct evidence is

shown to support different possibilities. Sentence also jumps from small-scale to large earthquake features

Line 10: referencing an in-preparation manuscript is bad practice and not appropriate and should be removed. Is there evidence to support difference in rheology to the matrix phases?

Line 15: how do you know that fluid was introduced? Needs some direct evidence

Lines 18-26: some of the fractures are accompanied by distinct Xalm and Xsps contents comparatively to the high-pressure rims, how is this accounted for under a high-pressure and not a low-pressure event. Presentation of modelling would aid this discussion

Line 29: add the, before the word most

Page 10 Line 1: what other possibilities could there be that could be discounted?

Line 3: some of these zones are barely present

Line 8: present the evidence to support this and discount the other possibilities, otherwise it feels a bit like special pleading

Line 31: this possibility should be discussed in light of other potential ways to resorb garnet, i.e. P-T paths involving consumption. This becomes impossible to evaluate as the modelling is not shown and it is unknown if water was added or subtracted from the modelling

Page 11 Line 10: best observed

Line 14: if this is re-equilibrating, what phases is it adjusting with? i.e. what is the volume and how can the P-T be robust if the equilibrium volume is a few microns within a garnet grain with introduced fluid

Line 7: the mineral equilibria modelling has not been shown and the paragenesis have not been outlined, these need to be discussed instead of sorely garnet chemistry

Line 10: jumps to the gross scale. A clear picture of the relationships of the different samples and their degree of change should be provided to aid the reader

Page 13 Line 24: this study does not show or discuss zircon, so how can we know it has brittle deformation features in its cores?

Line 33: if the diffusion of the different element influenced the re-equilibration volumes, then discussion and presentation of chemical potential gradients would aid the understanding of the potential re-equilibration processes

Figures Figure 1. If all samples are from the internal complex how can it be known what the difference in ingress occurred in the external complex?

Figure 2: should be all photomicrographs to discuss the texture then presentation of chemical maps. Additional photomicrographs of the general textures observed throughout the samples would provide better context to the garnet microstructure and evolution of the samples.

Figure 3: in many of the captions, interpretations are presented as facts. Interpretations should be avoided wherever possible in figure captions

Figure 6: avoid interpretations in the caption, especially surrounding re-equilibration it is not known what proportion of the rock volume is in equilibrium

Figure 8: where is the data from? How was it collected? if the data is from a previous publication this is inappropriate. What method was used to determine the P-T? what are the uncertainties? The high-pressure event overlaps suggesting any discrete events are not resolvable

Figure 9: This figure is inconsistent with Figure 8. How is rim1 formed at low-pressure but it is present as a high-pressure stage? Rim2 and rim3 are within error and likely

reflect one in the same process. The garnet modes have no context without the pseudosection, how is it know the water was introduced specifically at this time, was this incorporated into the model parameters?

I hope these comments help

---

## Referee Comment (RC2) · G. Ortolano (Referee) · 2 Oct 2017

The manuscript: "Deeply subducted continental fragments: I. Fracturing, dissolution-precipitation and diffusion processes recorded by garnet textures of the central Sesia Zone (Western Italian Alps)" by Giuntoli and coauthors is an interesting paper which deals with an alternative interpretation about the unravelling of a complex polyorogenic multistage history of an eclogite facies metasedimentary rocks from the Western Alps.

[Figure]

More in particular, a Variscan relic granulite facies paragenesis was replaced by an Alpine eclogite facies metamorphic cycle, which produced characteristics HP garnet overgrowth on previous HT garnet per each stage of the entire high dP/dT clockwise evolution. This process seems to be catalyzed by an initial brittle failure process, interpreted by the author as an early Alpine high strain rate deformational stage which brought to the formation of porphyroclastic texture, preserved in all of the observed garnet cores, as well as by a relatively thin mesh of microfractures sealed by grossular rich garnet. Although the work presents high-quality analytical data, unfortunately, the manuscript does not have a clear focusing line. It seems often a description of samples without sufficient context to allow the reader appropriate information to assess the processes proposed. The problem starts from the sample selection strategy. This is indeed not sufficiently justified to underline the specific peculiar features useful to better highlights the different mechanisms of garnet overgrowing stages developed during the Alpine evolution. It is out of sense for instance, during discussion, uses the name of the sample to describe the specific textural characteristics of the related Alpine garnet overgrowing stages. For an external reader, a name is a name. Instead, should be better associate a name to a specific process. Moreover, during in the introduction as well as in the discussion, were not taken into account any alternative possible interpretation for justifying the observed garnet texture. For instance, some brittle behavior can be generated not only by a high strain rate in non-coaxial regime but also by plastic-to brittle transition with the formation of a fractured mesh that might represent evidence of past episodic tremors or "slow earthquakes" triggered by high pore fluid pressure (Malatesta et al., 2017 Geological Magazine). What other evidence have the authors to justify their interpretation? Finally, in my opinion, the potentiality of the quantitative data extrapolation from image analysis by X-Map tools, was not satisfactory, in term for instance of the extrapolation of the effective reactant volumes of the single observed paragenetic equilibria. This can be useful to better constrain the ab initio parameters useful for a more consistent thermodynamic modeling, which unfortunately, was not described in the manuscript. For all of the above reasons, the

manuscript requires a deep major revision, consisting in a substantially rewriting of the introduction and of the discussion part, focusing the attention for instance to the use of the image analysis in the calculation of the effective bulk rock chemistries for the single extrapolation of paragenetic equilibria. Moreover, it is fundamental a better presentation of the complete methods utilized, together with a greater contextualization of sample selection. Specific details are reported in the attached pdf. Sincerely, Gaetano Ortolano.

Please also note the supplement to this comment:
https://www.solid-earth-discuss.net/se-2017-87/se-2017-87-RC2-supplement.pdf

**Supplement:**

[revised manuscript text omitted]

---

## Author Comment (AC1) · 1 Nov 2017

Below please find our comments to the **Anonymous Referee #1**, which are shown in Italic.
**Note:** at the end of this document, are available two new sections and a new figure that will be added in the revised manuscript, according to our replies to the two referees.

**Summary statement**
1. *Unfortunately, the data is presented without sufficient context to allow the reader appropriate information to assess the processes proposed. This renders the main conclusions regarding multiple Alpine metamorphic events unjustified, or extremely difficult to support based on the uncertainties within the data set. The manuscript requires the presentation of the complete methods utilized, ...*
We recognize that the two companion papers depend on each other, as well as two already published papers, for much of the data and methods used. In our revisions, we emphasize this interdependence and add specific pointers to the necessary information and its source. As for methods, we used in particular two programs (XMAPTOOLS 2.2.1 and GRTMOD). The approach was presented by Lanari et al. (2014, 2017); most of the details needed in the present manuscript are also discussed in these papers.
By adding a chapter at the outset, we expect that the doubts and difficulties this reviewer had in following our paper are now resolved. We also provide a summary on the methods used and the uncertainty estimation in the data set (chapter **5.2 Modeling phase equilibria in partially re-equilibrated rocks** in the revised version).
2. *... together with greater contextualization of the behaviour of other minerals within the rock to appropriately understand garnet and therefore eclogite evolution.*
The behaviour of the other minerals is presented extensively in the companion manuscript "Deeply subducted continental fragments: II. Insight from petrochronology in the central Sesia Zone (Western Italian Alps)", which is currently under review in Solid Earth and thus fully accessible. In the present manuscript, we specifically focus on garnet microtextures, thus adding this body of information again here would seem redundant.
3. *I understand that some of this information is presented in second installment or in different papers, if the data is reutilized it is not appropriate and leaves this manuscript with no novel quantitative information.*
The manuscript has an entire dataset that represent novel quantitative information not published elsewhere: the high-resolution compositional maps are based on well over two million EPMA analyses! The power of such maps to quantify the textural and microchemical archive in complex rocks is well established, so we are unsure what this reviewer is doubting here: The data we present are fully quantitative and entirely novel, i.e. they have not been presented elsewhere. We consider them the backbone to our modelling that lead us to unravel the fundamental processes – repeated growth and partial dissolution of garnet at high pressure conditions – and hence they are the core of the present paper.

**Specific comments**
4. *The main conclusions ascribing two periods of garnet growth/resorption during the high-pressure Alpine metamorphism is largely unconvincing.*
Further to the summary statement #3 we note that it is the combination of garnet textures and compositions presented, which indicate (at least) two discrete growth stages postdating the (pre-Alpine) core. Growth conditions (i.e. high pressures) were obtained by detailed thermodynamic modelling. Lobate textures and peninsular features inside the previous garnet generations revealed partial resorption, which in one sample also produced atoll garnet. The conspicuous microstructural evidence combined with the modelling results strongly indicate at least two high-pressure stages of garnet growth/resorption during Alpine time. We are unaware of any study ascribing comparable dissolution features to a process other than interaction with a reactive fluid.
5. *The P estimates established for rims2 and 3 are extremely similar, and within uncertainty estimates shown in Figure 8b. It is unknown how these estimates where determined outside of using a program GRTMOD making them hard to believe. What thermobarometry did the program use? What uncertainties are there on the method? What equilibrium volumes were assessed? This information together with the presentation and discussion of the complete thermodynamic models used, and their parameters is needed to justify the estimates provided. The presentation of pseudosections would also provide greater context to the history of the rocks.*
These questions showed us the need to give some details here. So, in our revised version, we introduced a chapter explaining the approach and details of the methods used, including the uncertainties in the data set. As stated in **5.2 Modeling phase equilibria in partially re-equilibrated rocks** in the revised version,

thermobarometry is based on isopleth intersection, and Lanari et al. (2017) explain how *GRTMOD* implements the estimates. Furthermore, our companion paper gives the details of garnet modeling (in supplement S8: GRTMOD results). Briefly, GRTMOD uses an inverse modeling approach; written in MATLAB©, it interacts with Theriak (de Capitani & Brown, 1987) using the Theriak_D (Duesterhoeft & de Capitani, 2013) extension. Specifically, for the samples presented here, the parameters chosen were given in **5.1.2 Garnet thermobarometry using GrtMod** section of the companion paper and also addressed in the revised version in the new chapter **5.2**. The equilibrium volume is assessed in a section of the companion paper (**5.2 Bulk rock and reactive bulk composition**), and P-T isochemical phase diagrams (pseudosections) are presented and discussed in that same paper (**5.3 Garnet thermobarometry and phase diagrams**).

6. *In particular, the questions surrounding what reactions are controlling garnet growth/resorption? What is the shape and habit of its isopleths? what part of the rock volume was equilibrated? These become difficult to assess in its current form, particularly by the assertion of fluid ingress. The influence on fluid needs to be discussed more fully in terms of the modelling volumes. If the equilibrium volumes are dependent on the diffusion of certain elements additional modelling of chemical potential gradients would be of potential benefit.*

(Please note that the first questions were partly addressed in the point above.) While textural and chemical evidence indicates that external fluid repeatedly interacted with nearly dry protoliths at HP conditions, we have no tight constraints on how much fluid entered at what spatial and temporal intervals. However, for each growth stage (within any one sample), the composition of garnet produced is uniform in each grain analysed, whereas the local geometry differs to some extent. This allows a spatial estimate of the reaction volume: Interaction of hydrous fluid with the reactive part of the assemblage (i.e. the matrix minerals) must have been at the scale of a thin section (centimetres) at least (except for the garnet relics left by incomplete reactions). Apart from garnet and local accessories (zircon, monazite), no mineral relics of the successive replacement reactions have been detected, indeed these rocks appear otherwise well equilibrated at eclogite facies conditions. Our modelling indicates that garnet resorption was dictated by changes in the PT conditions. As far as the suggestion to model diffusion based on the preserved chemical potential gradients in garnet, the complex and clearly diachronous overgrowth pattern would make this task extremely challenging, and we did not make such an attempt.

7. *In terms of the textural evolution of garnet, greater context of the surrounding phases is required to completely assess the rocks evolution, from both a microstructural and chemical perspective.*

We refer (as stated in the summary comments) to our companion paper, where the role of the other phases and their chemistry is extensively discussed.

8. *The sampling is from diverse compositions and protoliths, but consideration of the similarity of the processes is not adequately provided. It begs the explanation of how the period of brittle fracturing is linked to any remnant of current fabrics and their prevailing strain distribution locally and regionally? This manuscript largely states the processes of dissolution and precipitation without providing sufficient evidence to support and disprove possibilities. Presentation of quantitative microstructural information would greatly improve the submission and support some of the conclusions. Much of the focus is on garnet chemistry and not the rocks microstructure.*

These ambitions indeed have posed serious issues, all along the study presented here. Regrettably, the Sesia Zone presents limitations on them, and we may not have emphasized this sufficiently, though we address it repeatedly in the manuscript, e.g. on page 1, line 21: "Replacement of the Permian HT assemblages by hydrate-rich Alpine assemblages can reach nearly 100%" and on page 10, after line 23: "Apart from garnet and these accessory relics, the main pre-Alpine HT assemblage has completely re-equilibrated at eclogite facies conditions.".

Pre-Alpine or other pre-eclogite facies fabrics are otherwise hardly preserved in the study area. Specifically, the brittle stages visible – at micron scale – in garnet cores could not be directly linked to locally or regionally prevalent strain patterns. In the revised version, we shall emphasize these points some more, to clarify explicitly the limitations imposed by the polymetamorphic and polydeformed setting of this study.

It's not clear to us what "quantitative microstructural information" the reviewer would want to see. Our X-ray maps are based on quantitative data, and the information they contain is distinctly microstructural. Our study did not include PO fabric or grain size analysis etc., since we deem microchemical data to be far more relevant for the present purpose.

**Technical corrections**

*Technical corrections: Page 1 Line 20: "...Sesia Zone, with a general decrease in fluid-garnet interaction observed towards the external areas"*

Ok

*Line 22: 100% of what?*

Of the total, rephrased

*Line 23: what deformation structures? need to look at other phase than just garnet, and even then, you have only talked about brittle fractures. What about crystal plasticity?*

Here we meant deformation structures seen in the field, such as shear zones. We will rephrase this sentence to clarify what strain we address here.

*Page 2 Line 2: may alter the zonation recorded by garnet*

Ok

*Line 3: diffusion is also dependent on the medium or deformation i.e. grain boundary fluid or pipe diffusion*

Here we explicitly referred to intracrystalline diffusion in garnet.

*Paragraph 2: the introduction needs a clearer statement about what the manuscript is trying to solve*

Ok, we will strengthen this message (also in view of Ortolano's comments)

*Line 18: Permian high-temperature and Alpine high-pressure events. How have these events been determined in age?*

Here we refer to data in the published literature (references cited).

*Line 19: minerals shouldn't be pluralized*

Ok.

*Line 20: clumsy sentenced would benefit from rewording*

Agreed.

*Line 32: this is an interpretation and needs to be stated as such*

Ok, rephrased.

*Line 33: jumps to generalities of garnet growth from specifics of Alpine rocks, needs to be clearer and explicit following one logic flow then the next*

Good point, logic straightened.

*Page 3 Second paragraph: at this stage, it is not clear how you propose to evaluate these possibilities, and exclude different scenarios that lead to your preferred interpretation*

Ok, added a sentence in the revised version

*Line 9 and 10: confusing sentence needs rewording. As it reads it seems you collected the samples away from the foliation?*

Changed: We simply replaced "perpendicular to" by "across", which should avoid confusion.

*Line 11: delete single*

Ok

*Line 12: is the quantitative data already published? This is a major flaw and is not appropriate.*

In this sentence we referred just to the quantitative pressure and temperature data that are part of the companion paper. These data are used in this manuscript in relation with the microstructures, as we wrote. All the quantitative high-resolution compositional maps presented in this study are entirely novel. Please, see reply to summary statement 3

*Line 23: delete but*

Ok

*Page 4 Line 3: overprints*

Ok

*Line 4: granulite facies conditions and retrograde amphibolite facies conditions*

Ok

*Line 13: latter? Be explicit*

Ok, changed to "maps"

*Line 15 replace were with involved*

Ok

*Line 22: present the end member formulae used throughout*

Ok

*Line 24: ...representative amounts of the samples*

Ok

*Line 30: quantify size range*

It is from sub-millimetric to a few centimetres, we will add this information in the revised version

*Page 5 I think the use of rim doesn't really apply, they are interpreted growth zones. Using rim makes distinguishing the processes and stages difficult, as in most instance they are not physically rims. The microstructures, need more detailed descriptions and not left in the supplement. Grains sizes should be quantified. Requires photomicrographs*

Rim seemed the best term found to describe such growth zones, as most of them surround garnet cores. (Perhaps "seam" would be an alternative, but not clearly better suited.) Note that we specify "internal and external rims", where appropriate. (See pg.6, line 16 below.) We could indeed add a description regarding microtextures here, rather than in the supplement. No problem either to add a plate with photomicrographs showing the microstructures, if an additional figure is deemed necessary.

*Line 2: are these exsolution needles? If so this could be of significance to temperature or pressure estimates*

No, we think that these are fine inclusion of rutile that is abundant in our samples (1-2%) and in equilibrium with the eclogite facies fabrics. The origin or precise growth process remain uncertain.

*Line 6: from the internal and external areas*

Ok

*Line 23: sentence needs rewording*

Rephrased

*Line 30: I am not sure this is completely true, EBSD provides quantitative information about garnet microstructures, X-ray maps provide links between chemistry and textures. Deleting the word most would help this sentence.*

Ok, modified according to the comment

*Page 6 Line 1: change pictures to images*

Ok

*Line 10: How was the sealed fractures accounted for in the fractionated compositions and the modelling procedures?*

The sealed fractures have been not considered in modelling because of their small size and low abundance in terms of volume%. However, we observed that their composition matches that of Alpine rims (specifically of Rim2 in sample FG1315). Owing to the low modal abundance these fillings were not fractionated from the bulk rock to model the next growth zones. We can specify this in the revised version.

*Line 16: how is this accounted for? Growing in two directions*

In the revised version we will add some information and clarify the wording, as "Rim2 ($Alm_{64}Prp_{24}Grs_{11}Sps_1$) is found as three textural types: (a) It grew externally onto Rim1; (b) between core and Rim1, and also surrounds the Rim1 peninsula, thus extending it (by ~20 μm); (c) fine veins (5-20 μm thick) dissecting Rim1 also show Rim2 composition (as discussed in section 6.2). The outermost rim (Rim3) is lower in grossular ($Alm_{69}Prp_{24}Grs_6Sps_1$); it displays peninsular growths inside Rim1 and Rim2. Remarkably, Rim3 is thin (~100 μm) parallel to the main foliation and thicker (~400 μm) perpendicular to it."

*Line 19: providing some quantification of the growth scale in particular directions would strengthen conclusions, particularly across the diverse samples. However this does not explicitly consider crystallography*

See reply above

*Line 24: Almandine and spessartine vary a lot in the cores?*

Yes, this is shown in Fig. 5; and in the zoning profile of the garnet end-member in Fig. 2 of Lanari et al., 2017.

*Page 7 Line 2: the sealing of the fracture pattern is not obvious on the element maps?*

Yes it is: the compositional maps have a resolution high enough to display this change in chemistry between the garnet core and the garnet sealing the fracture pattern. Furthermore, and such difference is chemistry is constant in different areas of the garnet. A technical detail, the maps have step sizes between 3 and 5 micrometres and most of the fractures have a width that is at least three times major.

*Line 10: again, not obvious what this is talking about; the pyrope contents are unevenly distributed why is this so?*

This important observation is but described here; the interpretation and discussion are presented in **6.3 Re-equilibration close to fluid pathways**

*Line 17: remove interpretative statements in the results section*

Ok

*Paragraph 3 No results for the thermodynamic modelling are presented, this is more so thermobarometry. The models should be shown and the methods used to estimate P-T discussed. What parts of the garnet were fractionated (supplement figure maybe), how did this vary throughout the rest of the rock volume, garnet does not grow in isolation. No equilibration context is provided, it would be good to know the reactions controlling the growth/resorption. What are the nature of the garnet isopleths? Was fluid added and how did it change the modelling? These questions are not explicitly addressed. Discussion of the uncertainties of the P-T estimates are fundamental to resolving these discrete events, currently they are within errors presented in Figure 8 and cannot be resolved. The way this reads it seems the data has been presented elsewhere and therefore is largely re-interpreted and not appropriate as central conclusion of this manuscript, what is new here?*

Critique accepted. Please refer to the **Summary statement** and **Specific comments.** There we clarify that in this paper we link our microchemical and -textural data (presented in this manuscript) to our PT-data (presented in the companion manuscript). Whereas we are in doubt about the need to repeat details of the approach and models presented there, we do refer to the new paragraph introduced above.

*Page 8 Line 2: don't start a sentence with and*

Deleted "and"

*Line 7-14: there is an extremely large variation in pressure across all samples, encompassing the entire range of the high-pressure event, how can these be resolved as discrete metamorphic episodes?*

This is the range of all the P data, the single values are presented in Fig. 8. Each growth zone of garnet has a distinct chemical composition that suggests a series of metamorphic episodes related to changes in P,T and reactive bulk rock. In the area we studied, this is also reflected by several metamorphic and deformation stages occurring at eclogite facies conditions, such as the presence of a relic eclogite facies foliation preserved in microlithons and wrapped by the main eclogite facies foliation.

*Line 17: there is no U-Pb ages presented?*

No, but ages from the same samples set are presented in Kunz et al. (2017). Rephrased for the sake of clarity.

*Line 30: replace fillings with more indicative terminology*

Replaced with crack fillings

*Page 9 Line 1: veins? Keep terminology consistent*

OK, changed

*Line 4: best observed, delete visible. But less so for Mn*

Done

*Line 5: if concentration limits are near the lower limits of detection this should be discussed in the results*

Ok, added a sentence about it.

*Line 6: yes, grossular is commonly pressure dependent, but this should be show specifically for these samples together with the nature of garnet modes to assess potential resorption.*

See our reply to summary statement 1 and specific comment 5. In detail, this material is available in our companion paper (in supplement S8: GRTMOD results).

*Line 8: sentence needs restructure as it is confusing in its current form*

Rephrased

*Line 9: these options should be presented as possibilities unless direct evidence is shown to support different possibilities. Sentence also jumps from small-scale to large earthquake features*

This **is** presented as a possibility here ("may have been related…"); the jumps from small-scale to large earthquake features is what the reported studies advocated. We merely cite these studies as a possibility to explain the observed small-scale features.

*Line 10: referencing an in-preparation manuscript is bad practice and not appropriate and should be removed. Is there evidence to support difference in rheology to the matrix phases?*

At this stage this manuscript is still under review in $G^3$. We expect an editorial decision shortly and will update this reference or delete it if need be.

*Line 15: how do you know that fluid was introduced? Needs some direct evidence*

We consider the fracture networks in high-temperature garnet (cores) filled by high-pressure garnet as direct evidence. Less direct, but equally essential, is the volatile content of the rocks: Their pre-Alpine protoliths were granulites, i.e. essentially anhydrous rocks, whereas now they are micaschists containing up to 2 wt% $H_2O$ (see Table S1). If pre-Alpine rehydration had happened, e.g. say tectonic extension-related fluid had substantially hydrated the HT-protoliths, one would expect to see evidence of such as garnet alteration. Yet this is completely absent in all but one of our samples (see Fig. 9), in which a thin pre-Alpine (low-Ca) rim grew. Pre-Alpine hydration thus happened but locally and to a very limited extent. These lines of evidence jointly demand the introduction of substantial volumes of hydrous fluid, clearly in several episodes (on account of the different garnet rims), in the subducted slab.

*Lines 18-26: some of the fractures are accompanied by distinct Xalm and Xsps contents comparatively to the high-pressure rims, how is this accounted for under a high pressure and not a low-pressure event. Presentation of modelling would aid this discussion*

We disagree: The $X_{Alm}$ and $X_{Sps}$ in the sealed fractures are in the range of the high-pressure rim compositions we calculated by thermodynamic modelling. This is shown at page 6 lines 13, 27; page 7 lines 4, 14.

*Line 29: add the, before the word most*

Ok

*Page 10 Line 1: what other possibilities could there be that could be discounted?*

We did address these questions: On page 9 line 33 we stated "we surmise that Rim1 and 2 are not a simple growth sequence, with the older generation in a more internal position and younger ones more externally." We also discuss (and dismiss) the possibility that these growth features are related to a pressure cycling because we observe veins that resorbed Rim1 and connected the internal and external parts of Rim2. Perhaps these considerations are worth repeating in this part of the discussion (also in view of the comment below (line 8).

*Line 3: some of these zones are barely present*

A careful look at the figures shows that in samples FG12157 and FG1249 all zones are well visible.

*Line 8: present the evidence to support this and discount the other possibilities, otherwise it feels a bit like special pleading*

Ok, we can add this to the discussion in the revised version.

*Line 31: this possibility should be discussed in light of other potential ways to resorb garnet, i.e. P-T paths involving consumption. This becomes impossible to evaluate as the modelling is not shown and it is unknown if water was added or subtracted from the modelling*

Not clear what "this possibility" refers to. However, as modeling is a central topic in the companion paper, which discusses garnet resorption in response to changes in P-T, readers interested in evaluating the models will want to refer to that paper. We propose to point to the models, at this point in the revised manuscript.

*Page 11 Line 10: best observed*

Ok, changed

*Line 14: if this is re-equilibrating, what phases is it adjusting with? i.e. what is the volume and how can the P-T be robust if the equilibrium volume is a few microns within a garnet grain with introduced fluid*

Here we address intracrystalline re-equilibration inside the garnet core and specifically around the fractures. This situation differs from the equilibrium volume with a surrounding matrix. The reviewer's concern about PT-estimation is legitimate, but is it essential to explain the details here? The strategy used is fully explained in Lanari et al., 2017, as well as in the companion paper. We could state here that PT-estimates were based on the composition of unaffected areas of the core and the newly grown areas of the rims. However, some readers would probably find such detail to be out of context.

*Page 12*

*Line 7: the mineral equilibria modelling has not been shown and the paragenesis have not been outlined, these need to be discussed instead of sorely garnet chemistry*

These details and data are fully discussed in the companion paper. We will rephrase this sentence and add a reference of the companion paper

*Line 10: jumps to the gross scale. A clear picture of the relationships of the different samples and their degree of change should be provided to aid the reader*

This is extensively set out in the previous sections, but we probably should add a sentence here to summarize the spatial relation of the samples.

*Page 13 Line 24: this study does not show or discuss zircon, so how can we know it has brittle deformation features in its cores?*

This is correct. We propose to refer to a companion paper dealing with zircon, thus modifying the text (lines 23-25): In the sample suite reported here, pre-Alpine conditions are evident in garnet cores and relic zircon (Kunz et al. 2017). The relic features show brittle deformation textures, i.e. cracks, but no displacement.

*Line 33: if the diffusion of the different element influenced the re-equilibration volumes, then discussion and presentation of chemical potential gradients would aid the understanding of the potential re-equilibration processes*

Diffusion always influences the re-equilibration volume. A specific discussion in the present situation is outside the scope of the present paper and would be challenging given the complex zoning textures in the samples presented. Specifically, estimating the re-equilibration volume in the core is challenging, given the

high density of fractures, their complex 3D geometry, the repeated interaction with fluid at changing PT-conditions, etc.

***Figures***

*Figure 1. If all samples are from the internal complex how can it be known what the difference in ingress occurred in the external complex?*

This manuscript does not presented data for the External Complex, so implications on this unit are outside the scope of the present paper. We refer the reader to our companion paper for the geological evolution of the External Complex and its relations to the Internal Complex. The reason to omit the External Complex here is that it is composed mostly of orthogneiss that rarely contain garnet in the main paragenesis.

*Figure 2: should be all photomicrographs to discuss the texture then presentation of chemical maps. Additional photomicrographs of the general textures observed throughout the samples would provide better context to the garnet microstructure and evolution of the samples.*

Ok, good suggestion, we will add such photomicrographs (see at the bottom of this document)

*Figure 3: in many of the captions, interpretations are presented as facts. Interpretations should be avoided wherever possible in figure captions*

Ok, we will modify them according to this comment and indicate interpretations as such, where they are included.

*Figure 6: avoid interpretations in the caption, especially surrounding re-equilibration it is not known what proportion of the rock volume is in equilibrium*

Ok, as above.

*Figure 8: where is the data from? How was it collected? if the data is from a previous publication this is inappropriate. What method was used to determine the P-T? what are the uncertainties? The high-pressure event overlaps suggesting any discrete events are not resolvable*

As stated in the caption, we summarize relevant data from Lanari et al., 2017 and Giuntoli, 2016, so as to use them to distinguish Alpine from pre-Alpine growth zones. As stated above, we will introduce a brief section on modeling in this paper.

*Figure 9: This figure is inconsistent with Figure 8. How is rim1 formed at low-pressure but it is present as a high-pressure stage? Rim2 and rim3 are within error and likely reflect one in the same process. The garnet modes have no context without the pseudosection, how is it know the water was introduced specifically at this time, was this incorporated into the model parameters?*

The two figures actually are consistent: Sample FG1249 is the only one in which a pre-Alpine rim (pink square in Fig.8 a, b) was detected. We discuss this in the text and propose that Rim2 and Rim3 represent growths triggered by fluid influx, likely reflecting one and the same process. Select details about modeling will be introduced in the manuscript, as stated above, but for all of the information requested in this comment, the companion paper is indispensable.

We thank the Anonymous Referee #1 for so many constructive comments.

Francesco Giuntoli, Pierre Lanari, Martin Engi

**Below are available two new sections and a new figure that will be added in the revised manuscript, according to our replies to the two referees.**

New title of paragraph 4: "**4 Sample selection strategy and petrography**"

(Paragraph to be added after line 5 page 5)

These petrographic features match observations reported by previous authors (see references in section 1) who had attempted to distinguish Alpine form pre-Alpine garnet in the Sesia Zone. In light of these observations, the sample selection strategy we adopted for this study used the following criteria:

- The presence in hand-specimens of garnet of diverse sizes (sub-millimetric to a few centimetres);
- The bimodal distribution of garnet in one and the same thin section, with the larger crystals displaying a bright core and dark rims, as illustrated above, and smaller euhedral crystals (Fig. 2);
- Bright cores surrounded by darker rims evident in the SEM with a BSE detector.

This list of criteria was adopted to investigate the widest possible range of microtextures and processes recorded by garnet. Applying the list lead us to concentrate on only ~5% of the total samples taken, mostly because larger (pre-Alpine) garnet grains rarely survived in the area of study. In fact, almost all of the pre-Alpine HT assemblages had re-equilibrated in hydrate-rich Alpine assemblages.

[Figure]

**Figure 2: Thin section optical microphotos displaying the eclogitic foliation ($S_{ecl}$, red dashed line) and the bimodal size range of garnet (large garnet grains Grt-L; small garnet grains Grt-S; see text for further details). Black squares indicate the location of the high-resolution X-ray maps. (a) $S_{ecl}$ wrapping Grt-L grains in sample FG1315. (b) Same thin section as (a) with atoll garnets (Grt-S) located within a Qz rich band and a Ph rich band. (c) and (d) Folded $S_{ecl}$ marked by Ph, Gln and Rt; Chl is present in the fold hinges. Plane-polarized light: (a, b, c); cross-polarized light (d).**

**5.2 Modeling phase equilibria in partially re-equilibrated rocks**

In each sample investigated, several garnet growth zones were identified by careful analysis of the end-member proportion maps using XMAPTOOLS (Lanari et al., 2014). For modelling, representative areas of each growth zone were extracted from this dataset, using the program's export function to obtain average chemical compositions. The quantitative micro-mapping strategy employed in this study has well established advantages (e.g. Marmo et al., 2002;Lanari et al., 2013;Ortolano et al., 2014;Angiboust et al., 2016) over traditional spot analyses: (1) it allows key relationships, such as the successive growth zones, to be identified and relevant compositions to be constrained, (2) it permits testing if chemical zoning patterns are consistent over several grains, which helps support (or refute) the assumption of grain boundary equilibrium, (3) it can be used to approximate local reactive bulk composition by accounting for mineral relics. In all the samples of the present study, the growth zone patterns and compositions of the mineralogical phases were consistent at thin section scale.

The complexity of the garnet compositional zoning shown in Figures 3, 4, 5, 6 indicates that isochemical phase diagrams (or pseudosections) must be used with due caution. Previous studies have demonstrated that garnet fractionation can sensibly affect the reactive bulk composition (Evans, 2004;Robyr et al., 2014;Konrad-Schmolke et al., 2008) and thus shift the calculated garnet isopleths in a P-T diagram (Lanari and Engi, 2017). However, garnet fractionation is not easy to account for where several growth stages are evident, as well as intermittent dissolution (which we show to be the case in the companion paper by Giuntoli et al., submitted). Since the older growth zones are but partially preserved, as indicated e.g. by lobate edges (Figs. 2, 3, 5, 6) in our samples, a novel strategy was developed (Lanari et al., 2017). It relies on the optimization of the reactive bulk composition, and the computer code (GRTMOD) presented in that study was applied to the present samples. In essence, for each garnet growth zone, the reactive bulk composition was optimized jointly with the P-T conditions to predict (using Theriak-Domino, de Capitani and Petrakakis, 2010) a garnet composition that matched the measured one. Results were accepted if the residual value (the sum of the fraction of end-members) between the modelled and observed garnet compositions was <0.05, reflecting a close match. Previously formed garnet generations were removed from the bulk rock composition, according to the end-member proportion maps analysis. Analogously, in the case of garnet resorption, the appropriate components were again added to the reactive bulk composition. This iterative modeling was applied to each successive growth zone. The resulting P-T estimates are reported in Figure 8b, with error bars showing the P-T uncertainty related to the analytical error of the garnet composition (Lanari et al., 2017). For any given reactive bulk composition, narrowly spaced isopleths return small uncertainty envelopes, whereas widely spread isopleths return larger ones. A detailed account of these methods and the internally consistent thermodynamic dataset used, including solid solution models, is presented in the companion paper (Giuntoli et al., submitted).

Here follows **5.3 Results of thermodynamic modelling of garnet growth zones** (page 7 line 20 of the submitted manuscript), previously number 5.2

**References**

Angiboust, S., Yamato, P., Hertgen, S., Hyppolito, T., Bebout, G., and Morales, L.: Fluid pathways and high pressure metasomatism in a subducted continental slice (Mt. Emilius klippe, W. Alps), J. Metamorph. Geol., 2016.

de Capitani, C., and Petrakakis, K.: The computation of equilibrium assemblage diagrams with Theriak/Domino software, Am. Mineral., 95, 1006-1016, 10.2138/am.2010.3354, 2010.

Evans, T. P.: A method for calculating effective bulk composition modification due to crystal fractionation in garnet-bearing schist; implications for isopleth thermobarometry J. Metamorph. Geol., 22, 547-557, 2004.

Giuntoli, F., Lanari, P., Burn, M., Kunz, B. E., and Engi, M.: Deeply subducted continental fragments: II. Insight from petrochronology in the central Sesia Zone (Western Italian Alps), Solid Earth, submitted.

Konrad-Schmolke, M., O'Brien, P. J., De Capitani, C., and Carswell, D. A.: Garnet growth at high- and ultra-high pressure conditions and the effect of element fractionation on mineral modes and composition, Lithos, 103, 309-332, 2008.

Lanari, P., Riel, N., Guillot, S., Vidal, O., Schwartz, S., Pêcher, A., and Hattori, K. H.: Deciphering high-pressure metamorphism in collisional context using microprobe mapping methods: Application to the Stak eclogitic massif (northwest Himalaya), Geology, 41, 111-114, 2013.

Lanari, P., Vidal, O., Lewin, E., Dubacq, B., De Andrade, V., and Schwartz, S.: XMapTools a Matlab©-based graphic user interface for microprobe quantified image processing, Computers and Geosciences, 62, 227-240, 10.1016/j.cageo.2013.08.010, 2014.

Lanari, P., and Engi, M.: Local Bulk Composition Effects on Metamorphic Mineral Assemblages, Reviews in Mineralogy & Geochemistry, 83, 55–102, http://dx.doi.org/10.2138/rmg.2017.83.1, 2017.

Lanari, P., Giuntoli, F., Loury, C., Burn, M., and Engi, M.: An inverse modeling approach to obtain P-T conditions of metamorphic stages involving garnet growth and resorption, Eur. J. Mineral., 29, 181-199, 10.1127/ejm/2017/0029-2597, 2017.

Marmo, B., Clarke, G., and Powell, R.: Fractionation of bulk rock composition due to porphyroblast growth: effects on eclogite facies mineral equilibria, Pam Peninsula, New Caledonia, J. Metamorph. Geol., 20, 151-165, 2002.

Ortolano, G., Zappalà, L., and Mazzoleni, P.: X-Ray Map Analyser: A new ArcGIS® based tool for the quantitative statistical data handling of X-ray maps (Geo-and material-science applications), Comput. Geosci., 72, 49-64, 2014.

Robyr, M., Darbellay, B., and Baumgartner, L. P.: Matrix-dependent garnet growth in polymetamorphic rocks of the Sesia zone, Italian Alps, J. Metamorph. Geol., 32, 3-24, 2014.

---

## Author Comment (AC2) · 1 Nov 2017

Here we reply to the comments of **Ortolano-Referee #2**. We reported in Italic the Ortolano-Referee #2 text. Note: at the end of the document "Reply to Anonymous Referee1" are available two new sections and a new figure that will be added in the revised manuscript, according to our replies to the two referees.

**Summary statement**

- *Although the work presents high-quality analytical data, unfortunately, the manuscript does not have a clear focusing line. It seems often a description of samples without sufficient context to allow the reader appropriate information to assess the processes proposed. The problem starts from the sample selection strategy. This is indeed not sufficiently justified to underline the specific peculiar features useful to better highlights the different mechanisms of garnet overgrowing stages developed during the Alpine evolution.*

Thanks for these suggestions: We will strengthen the Introduction, stating more clearly the specific goals of this paper. In the revised version, we will also change the name of Section 4 in "**4 Sample selection strategy and petrography**", we shall include a paragraph on important aspects of our sample evaluation strategy.

- *It is out of sense for instance, during discussion, uses the name of the sample to describe the specific textural characteristics of the related Alpine garnet overgrowing stages. For an external reader, a name is a name. Instead, should be better associate a name to a specific process.*

We understand that the sample names we use have no meaning to an external reader, but we think that keeping them in the text is important and fundamental to be able to discuss the different processes in the light of the garnet textures. Such textures have some similarities amongst samples but also remarkable peculiarities and differences. Without referring to the sample names, specific observation and discussion would result hard or impossible to follow to an external reader. It would also diminish the possibility of the reader to self-asset the interpretation of such textures and it would render the discussion section much more subjective to the eyes of an external reader. Furthermore, to give more result to the processes and less to the samples names, we already grouped the different processes as is visible from the different subsection of section 6 (**6.1 Micrometre-size fracture network in garnet cores, 6.2 Resorption and growth: fluid-related textures, 6.3 Re-equilibration close to fluid pathways**) and discussed similarities amongst samples for each of these. Finally, this sample series is part of other publications; some of these are already published and have this sample nomenclature: changing it would results to confusion and would damp comparison of the data.

- *Moreover, during in the introduction as well as in the discussion, were not taken into account any alternative possible interpretation for justifying the observed garnet texture. For instance, some brittle behavior can be generated not only by a high strain rate in non-coaxial regime but also by plastic-to brittle transition with the formation of a fractured mesh that might represent evidence of past episodic tremors or "slow earthquakes" triggered by high pore fluid pressure (Malatesta et al., 2017 Geological Magazine). What other evidence have the authors to justify their interpretation?*

This comment seems partially unfounded: In the Introduction, we outline three possible interpretations of the garnet textures reported in the literature from the Sesia Zone (page 2 lines 19-32); alternative processes leading to the formation of atoll garnet are presented (page 2 line 32, page 3 line 4). Furthermore, we propose two alternatives processes to account for the development of fractures (page9 lines 15-24). For reasons outlined in section 6.1 we favour the first interpretation. It is correct, however, that we can improve the discussion section by stressing which data and textures support or confute alternative interpretations (as referee#1 also commented).

Regarding the specific suggestion *(following Malatesta et al., 2017)* of *tremors or "slow earthquakes" triggered by high pore fluid pressure* possibly leading to brittle behaviour, we think that this situation applies to our dataset: The interpretation by *Malatesta et al.* is based on lithotypes with strong rheological contrast, i.e. metasediments alternating with metabasites, separated by cm-thick talcschist layers, so metabasite shows brittle fracturing (boudinage, brecciation) inside the weak matrix (Fig.4, 14). In our samples, brittle fractures are observed in garnet (and zircon) cores, which are relics of a dry granulite that must have been mechanically strong. We see strong analogies in our situation with the fracture patterns and compositional maps of garnet reported by Austrheim et al., 1996; Angiboust et al., 2012 and Austrheim et al., 2017. These authors interpreted such textures as produced by high strain rates related to seismic failure. For this reason we tentatively adopt such an interpretation to explain the features we observed in garnet (page 9 lines 9-11).

- *Finally, in my opinion, the potentiality of the quantitative data extrapolation from image analysis by X-Map tools, was not satisfactory, in term for instance of the extrapolation of the effective reactant volumes of the single observed paragenetic equilibria. This can be useful to better constrain the ab initio*

*parameters useful for a more consistent thermodynamic modeling, which unfortunately, was not described in the manuscript.*

As in our reply to Anonymous Referee #1 comments, these data are part of the companion paper "Deeply subducted continental fragments: II. Insight from petrochronology in the central Sesia Zone (Western Italian Alps)" currently under review in Solid Earth and fully accessible. We agree that these data are necessary to support our interpretations, but the volume and diversity of material is such that we decided to present it in companion papers. In the present manuscript, wherever data are particularly critical for our interpretation, we refer the companion paper (as Giuntoli et al., submitted). However, since both Ortolano and Referee#1 had difficulties to see the connection and asked for some clarification on methods we used, a section (**5.2 Modeling phase equilibria in partially re-equilibrated rocks**) will be introduced in our revised version to explain our approach to thermodynamic modeling of garnet.

- *For all of the above reasons, the manuscript requires a deep major revision, consisting in a substantially rewriting of the introduction and of the discussion part, focusing the attention for instance to the use of the image analysis in the calculation of the effective bulk rock chemistries for the single extrapolation of paragenetic equilibria. Moreover, it is fundamental a better presentation of the complete methods utilized, together with a greater contextualization of sample selection.*

We propose to improve the Introduction according to Ortolano's comments. For the use of image analysis in approximate effective bulk rock compositions from local composition, but we do not believe that this is necessary in this paper. The bulk rock composition was used for modelling, this is quite common and the good results of the models partially justify this assumption. The reproducibility of the zoning pattern at the centimetre scale also supports this assumption. It is thus not necessary to define a smaller equilibration volume that is also not supported from a textural point of view. The garnet composition is for instance the same in the phyllosilicate-rich layers and in the quartz-rich layers supporting the grain boundary equilibrium model assumed here. All these points are already discussed in some details in Lanari et al. (2017), presenting the GRTMOD program. A new modelling section with some computational details will be introduced in the revised manuscript (see previous comment). As stated above, we shall also add a section regarding the sample selection (collection and evaluation) strategy we used.

**Specific comments (from Ortolano's pdf supplement file)**

- *It is a zoning or an overgrowing crystallization*

To avoid misinterpretation at this stage, we prefer the "zoning" as a purely descriptive term.

- *Local texture and mineral chemistry are combined to define the ab inition constraints for a more consistent thermodynamic modelling. This last is function of the textural and mineral chemical features of the specific paragenetic equilibria*

Rephrased according to this comment.

- *Please introduce the sample selection logic, before to describe the chracteristic of the single sample. The sentence is not so clear, please rewrite.*

This is good advice and, as stated in the summary statement, we will adopt it.

- *This is the unique solution?*

See the above discussion in summary statement

- *Please emphasize that the thermodynamic modelling was assisted by quantitative image analysis useful to extrapolate the effective bulk rock chemistries of each paragenetic equilibria*

Ok, we introduce some details about modeling in the main text, as stated above

- *A robust thermodynamic modelling derives from a quantitative extrapolation of the effective reactant volumes of the single metamorphic evolutionary stage*

We present this topic in the new section about modeling

- *How many samples with what logical selection*

As above, we introduce this in the section on sample selection strategy

- *This is an anticipation of the discussion. Please avoid it.*

Ok, deleted.

- *How many thin sections. What is the logic of sample collection and more in particular, what is the logic of sample selection of those sample used for the thermodynamic modelling?*

We add this in the section on sample selection strategy

- *Please specifies better the logical meaning of the proposed procedure and the specific results that the authors want to reach for the aims of the present work.*

Ok, we will add such details, as well in the new section **5.2 Modeling phase equilibria in partially re-equilibrated rocks** that will be added in the revised version.

- *Also in this case, please specifies the logical meaning at the base of the use of XMapTools, such for instance to unravel the effective bulk rock chemistries of the modelled systems and so on.*

This will be presented in the new section **5.2 Modeling phase equilibria in partially re-equilibrated rocks** that will be added in the revised version.

- *Please specifies the logical process of the sample collection campaign and the following logical meaning of selection of those samples considered characteristic of....(e.g. Alpine prograde metamorphism; Retrograde Variscan metamorphism and so on...)*

Yes, we introduce this in the section on sample selection strategy.

- *More than a compositional zoning, I would talk about Evolution of the garnet overgrowing stages*

We consider this suggestion as a good alternative

- *This fractures are very intersting. Just for suggestion, if you use the X-ray Map Analyser (Ortolano et al., 2014 C&G), you can probably extrapolate the specific principal component of the classified image which correspond to the different generation of sealed fractures.*

That's correct, and in fact we followed this suggestion but using XMapTools; the results are presented in section 5 (e.g. page 6 line 13). A clarification: we see one generation of fractures sealed by garnet, a second generation is cutting across all of the garnet growth zones with chlorite lining. This is related to the retrograde greenschist stage, a late metamorphic phase evident in many parts of the Sesia Zone.

- *Principal Component Analysis indeed can highlight the specific interdependence of the different elemental components, emphasizing the presence of specific subphase, using the second analytical cycle of X-Ray Map Analyser.*

Ok this is true, but it can also be achieved in a simple binary chemical diagram, as only two chemical variables are independent in this case (XAlm and XGrs for instance with XPrp being dependent).

- *It is out of sense indicate a subparagraph of the manuscript with a samèle name. An external reader would understand the specific significance of that sample.*

Not sure we understand this comment. This hierarchy in fact aims to help readers to follow the presentation of the data. Possibly we could change a (sub)heading.

- *Where is the thermodynamic modelling approach. How it was calculated the Effective Bulk Rock chemistry of each garnet overgrowing stage.*

See our summary statement: A summary of the approach will be added to the revised manuscript. Note that the effective bulk composition is part of the optimization function to be able to predict fractionation or resorption (see Fig. 3 and 4 in Lanari et al. 2017)

- *Discussion have to be rewrite to better focus the aims of the paper, taking into account previous or potential different interpretations, supporting the present one with more consistence.*

The revised Discussion will take this suggestion into account (in line with the comments of Referee #1).

- *The shape of the boundary for the study area identification not seem to be the same of the Fig. b.*

The shape of the study area is not exactly the same in the two maps due to graphic reasons and the big difference in the map scales, nonetheless it is representative

- *What is c*

Mistake corrected

- *To thick*

Ok, reduced

- *This image should be better emphasized with the use of the principal compoent analysis*

See our previous reply to the specific comment on "*Principal Component Analysis*"

- *These images should be better emphasized with the use of the principal compoent analysis*

See our previous reply to the specific comment on "*Principal Component Analysis*"

- *This figure look very good*

Thanks

We thank G. Ortolano-Referee #2 for his constructive comments.

Francesco Giuntoli, Pierre Lanari, Martin Engi